# Anti-viral defence by an mRNA ADP-ribosyltransferase that blocks translation

Christopher N. Vassallo[1,3], Christopher R. Doering[1,3] & Michael T. Laub[1,2 ✉]

Host–pathogen conflicts are crucibles of molecular innovation[1,2]. Selection for immunity to pathogens has driven the evolution of sophisticated immunity mechanisms throughout biology, including in bacterial defence against bacteriophages[3]. Here we characterize the widely distributed anti-phage defence system CmdTAC, which provides robust defence against infection by the T-even family of phages[4]. Our results support a model in which CmdC detects infection by sensing viral capsid proteins, ultimately leading to the activation of a toxic ADP-ribosyltransferase effector protein, CmdT. We show that newly synthesized capsid protein triggers dissociation of the chaperone CmdC from the CmdTAC complex, leading to destabilization and degradation of the antitoxin CmdA, with consequent liberation of the CmdT ADP-ribosyltransferase. Notably, CmdT does not target a protein, DNA or structured RNA, the known targets of other ADP-ribosyltransferases. Instead, CmdT modifies the N6 position of adenine in GA dinucleotides within single-stranded RNAs, leading to arrest of mRNA translation and inhibition of viral replication. Our work reveals a novel mechanism of anti-viral defence and a previously unknown but broadly distributed class of ADP-ribosyltransferases that target mRNA.

ADP-ribosyltransferases are important enzymes found throughout biology that transfer the ADP ribose moiety of $NAD^+$ onto other biomolecules, usually modifying an amino acid on a target protein[5–7]. ADP-ribosylation is one of the most common post-translational modifications in biology, and is used to regulate a wide variety of proteins involved in cellular signalling, chromatin and transcription, DNA repair, and other functions. At least six of the human ADP-ribosyltransferases (previously called poly-ADP-ribosyl polymerases[8] (PARPs)) are induced by interferon and are thus hypothesized to function in anti-viral defence[9,10], but their precise roles and targets remain poorly defined. Although they have been studied for decades as protein modifiers, recent work has identified ADP-ribosyltransferases that target nucleic acids. DarT toxins of DarTG toxin–antitoxin systems can modify single-stranded DNA[11]. The *Pseudomonas aeruginosa* type VI secretion effector RhsP2 targets the 2′ hydroxyl of some double-stranded RNAs[12], and in *Photorhabdus laumondii*, a type VI secretion effector modifies 23S ribosomal RNA[13] (rRNA).

Here, we identify CmdT, an ADP-ribosyltransferase that functions in bacterial anti-phage defence by specifically modifying mRNAs to block phage translation and the production of mature virions. CmdT is part of a tripartite toxin–antitoxin–chaperone (TAC) system. Toxin–antitoxin systems have a major role in anti-phage defence[14]. These systems feature a protein toxin that is restrained from killing a cell or blocking cell growth by a cognate antitoxin[15]. For anti-phage toxin–antitoxin systems, phage infection must somehow liberate the toxin, but the mechanisms responsible remain incompletely understood[16]. TAC systems are common variants that feature a chaperone related

to the conserved protein export chaperone SecB. The chaperones of TAC systems promote proteolytic stability of their cognate antitoxin and, consequently, neutralization of the toxin[17–20]. How the toxins of TAC systems are liberated is not known.

For CmdTAC from the *Escherichia coli* strain ECOR22[4], we find that during infection by T4 phage, the major capsid protein outcompetes the antitoxin CmdA for binding to the chaperone CmdC. Consequently, CmdA is degraded by the ClpP protease, leading to activation of the CmdT ADP-ribosyltransferase. Notably, CmdT does not target proteins, DNA or structured RNAs, the targets of other known ADP-ribosyltransferases. Instead, we find that CmdT selectively modifies mRNA to block translation and abort the phage infection. Biochemical analyses demonstrate that CmdT primarily modifies the N6 position of adenine in GA dinucleotides found in mRNAs. In sum, our work reveals a novel mechanism of anti-phage defence and a previously unknown molecular target of an ADP-ribosyltransferase. As some of the interferon-inducible human ADP-ribosyltransferases are reported to also modify RNA in vitro[10], our work suggests that ADP-ribosylation of RNA may be a common, previously unappreciated facet of innate immunity throughout biology.

## CmdTAC protects against *Tevenirinae* phages

The *cmdTAC* system (Fig. 1a) was discovered in the genome of the wild *E. coli* isolate ECOR22 through a functional genetic screen for anti-phage defence systems[4]. Its homologues are found in diverse bacteria, including both Gram-negative and Gram-positive species (Extended Data

[1]Department of Biology, Massachusetts Institute of Technology, Cambridge, MA, USA. [2]Howard Hughes Medical Institute, Cambridge, MA, USA. [3]These authors contributed equally: Christopher N. Vassallo, Christopher R. Doering. ✉e-mail: laub@mit.edu

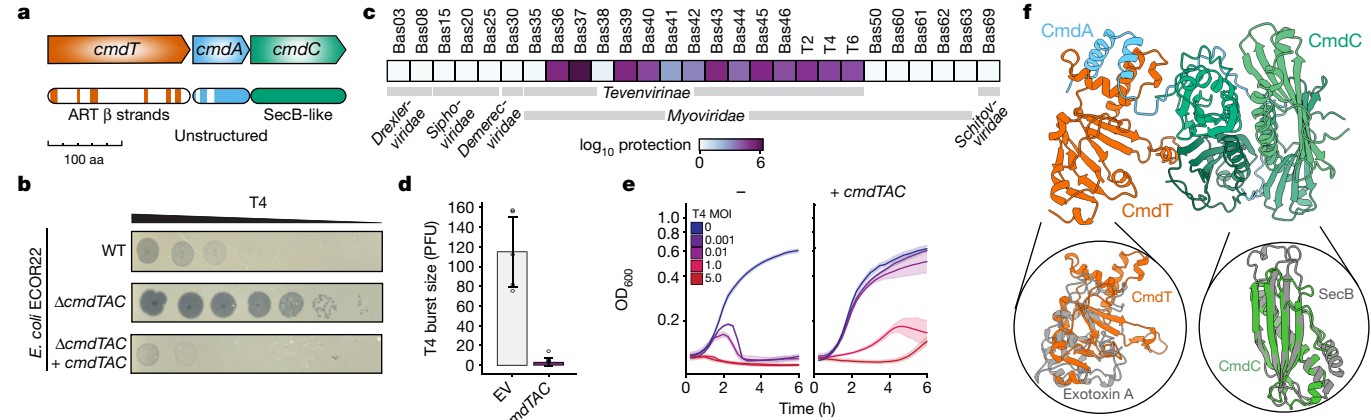

**Fig. 1 | The anti-phage defence system CmdTAC protects *E. coli* against infection by *Tevenvirinae*. a**, Schematic of the *cmdTAC* operon (NCBI proteins RCP66309-11.1) indicating structural features predicted by HHpred. ART, ADP-ribosyltransferase. **b**, Plaquing of tenfold serially diluted T4 phage on lawns of *E. coli* strain ECOR22 and an ECOR22 Δ*cmdTAC* mutant without or with a low-copy plasmid containing *cmdTAC* under its native promoter. **c**, Efficiency of plating of the phages indicated on *E. coli* K-12 + *cmdTAC* relative to an empty vector control. T2, T4 and T6 data from ref. 4. **d**, T4 burst size in *E. coli* K-12 + *cmdTAC* or an empty vector (EV) (1 h at 30 °C). Data are mean ± s.d.; *n* = 5

biological replicates. **e**, Growth of *E. coli* K-12 + *cmdTAC* or empty vector and infected with T4 at the indicated MOIs. The centre line indicates mean, shaded area represents 95% confidence interval; *n* = 6 technical replicates; representative of 3 biological replicates. **f**, AlphaFold2-predicted model of CmdTAC complex with insets showing structural alignments of CmdT to the ADP-ribosyltransferase exotoxin A (Protein Data Bank (PDB) 1AER, root mean square deviation (r.m.s.d.) 6.93 Å) and CmdC monomer to the SecB chaperone monomer (PBD 1OZB, r.m.s.d. 4.13 Å). r.m.s.d. values were generated by Foldseek.

Fig. 1a,b). In *E. coli*, *cmdTAC* provides robust defence against the *Tevenvirinae*, a major family of phages, which includes T4. Consistent with our prior work[4], deleting *cmdTAC* from ECOR22 improved the efficiency of plating (EOP) of T4 by at least 10³-fold. Transformation of ECOR22 with a low-copy plasmid containing *cmdTAC* expressed from its native promoter (hereafter *cmdTAC* refers to this construct unless noted) fully restored defence (Fig. 1b). We challenged *E. coli* MG1655 containing *cmdTAC* with all 12 *Myoviridae* subfamily *Tevenirinae* phage from the BASEL collection[21]. CmdTAC decreased the EOP of most *Tevenirinae* phages between 10²- and 10⁵-fold with only two exceptions, Bas35 and Bas38 (Fig. 1c and Extended Data Fig. 1c). No protection was observed against phages from other taxa. We also measured burst size during a single round of infection and found that CmdTAC reduced the number of T4 progeny from around 110 phages to almost 0, indicating a lack of new viral particle production (Fig. 1d).

To determine how CmdTAC provides defence, we examined growth of *E. coli* MG1655 containing *cmdTAC* following infection by T4 at different multiplicities of infection (MOI) (Fig. 1e). Direct defence systems provide protection at almost any MOI, whereas abortive infection (Abi) systems allow growth only when the MOI is less than 1. Abi systems typically inhibit the growth of infected cells to prevent phage replication and thus do not protect the individual cell, but prevent the spread of the infection to other cells in the population[22]. Indeed, *cmdTAC* protected against infection only at MOIs less than 1, with the survival of infected cells indistinguishable from an empty vector control (Extended Data Fig. 1e). By contrast, around 15% of control cells with a direct defence system (PD-T4-1[4]) survived. We also found that the number of infected cells that produced at least 1 progeny phage (that is, the efficiency of centre of infection (ECOI)) was reduced nearly 100-fold relative to cells lacking *cmdTAC* (Extended Data Fig. 1f). These results are consistent with CmdTAC functioning via an Abi mechanism.

## CmdA helps fold and neutralize CmdT

To identify the potential functions of each component in the CmdTAC system, we used HHpred and AlphaFold2 together with FoldSeek to predict the function and structure of each component separately and in complex (Fig. 1f and Extended Data Fig. 2a–f). These analyses identified ADP-ribosyltransferase-like β-strands in CmdT and predicted

a structure with similarity to the catalytic domain of *P. aeruginosa* exotoxin A, an ADP-ribosyltransferase of the diphtheria toxin (DTX) family[23]. A sequence logo built from CmdT homologues revealed several conserved residues including the HY-[E,D,Q] catalytic triad of the diphtheria toxin family of ADP-ribosyltransferases[24] (Extended Data Fig. 3a). Substituting alanine for the inferred catalytic residue of CmdT, Y41—a variant referred to here as CmdT*—abolished protection against T4 (Extended Data Fig. 1d). CmdA was predicted to have a pair of N-terminal α-helices followed by a long and unstructured C-terminal domain[20]. CmdC is homologous to SecB and has the same predicted tetrameric organization as SecB-like chaperones commonly found in TAC systems[18,19]. AlphaFold2 predicted that the three proteins form a hetero-hexameric complex with a 1:1:4 ratio of CmdT:CmdA:CmdC (Fig. 1f). To validate that CmdTAC forms a complex, we engineered *cmdTAC* under its native promoter to produce CmdT–Flag and then performed immunoprecipitation coupled to tandem mass spectrometry (IP–MS/MS). CmdT co-precipitated with both CmdA and CmdC (Extended Data Fig. 3b and Supplementary Table 1).

To experimentally test their predicted functions, we expressed each gene of the *cmdTAC* system in *E. coli* MG1655. Inducing any individual component did not strongly affect plating viability (Fig. 2a). However, co-expressing *cmdT* and *cmdA* led to a substantial loss in viability. This phenotype was abolished by the Y41A substitution in the putative catalytic domain of CmdT, indicating that CmdT is likely to be a toxic ADP-ribosyltransferase (Fig. 2a). Producing CmdT alone was not toxic, probably because the protein did not steadily accumulate, as measured by SDS–PAGE, and was insoluble, as indicated by a complete lack of protein detected in native gels (Fig. 2b and Extended Data Fig. 3c). By contrast, CmdT co-produced with CmdA steadily accumulated in both SDS–PAGE and native gel analysis. We then repeated the *cmdTA* co-expression experiment, but with CmdA containing a 3×HA epitope, and found that CmdA transiently accumulated and then was almost undetectable after 75 min of induction (Fig. 2c). Together, our results indicate that when co-expressing *cmdTA*, CmdA is initially needed for the folding or stabilization of CmdT, but that CmdA is subsequently cleared from cells, leaving behind stable, toxic CmdT.

Similar to the antitoxins of other known TAC systems, the C terminus of CmdA ends with LAA, a ClpXP protease recognition sequence[25]. To determine whether CmdA is degraded by ClpP, we induced expression

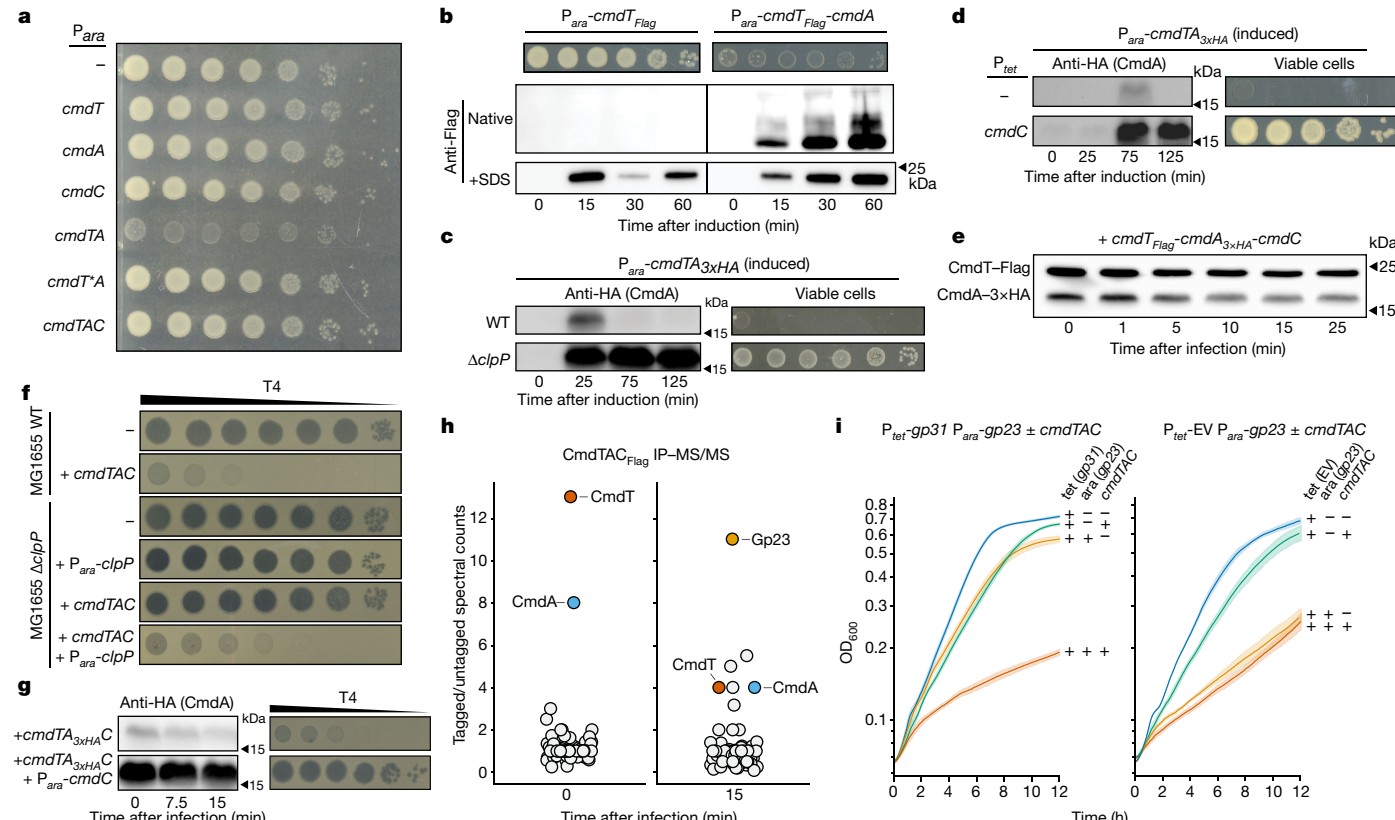

**Fig. 2 | CmdTAC is a TAC system that is activated by the T4 major capsid protein. a**, Plating viability (tenfold serial dilutions) of strains containing empty vector or the indicated components and induced with 0.2% arabinose from an arabinose-inducible promoter (P$_{ara}$). **b**, Cells expressing *cmdT* or *cmdTA* with a Flag tag on the C terminus of CmdT were assessed for plating viability by growth on LB + arabinose (top) and CmdT expression by immunoblotting of native gels (middle) or SDS–PAGE (bottom). Loading control shown in Extended Data Fig. 3c. **c**, MG1655 Wild-type or MG1655 Δ*clpP* cells expressing *cmdTA* with a 3×HA tag at the N terminus of CmdA were assessed for CmdA levels by immunoblotting (left) and plating viability on LB + arabinose (right). **d**, Cells expressing *cmdTA* with CmdA harbouring an N-terminal 3×HA tag. The cells also contain empty vector or a vector with *cmdC* under the control of a tetracycline-inducible promoter. CmdA levels were measured by immunoblot (left) and cell viability was assessed by tenfold

serial dilutions on LB with inducers (right). **e**, Immunoblots of cells harbouring *cmdTAC* with CmdT and CmdA engineered to have a Flag or 3×HA tag on their C and N terminus, respectively. Samples taken from cells infected with T4. Loading control shown in Extended Data Fig. 3e. **f**, Phage plaques of tenfold serially diluted T4 on MG1655 wild-type or MG1655 Δ*clpP E. coli* with empty vector or the indicated plasmids. **g**, Cells with *cmdTAC* (with N-terminally 3×HA tagged CmdA) and expressing an additional copy of *cmdC* (bottom) or not (top) were assessed for CmdA levels by immunoblot (left) and for defence by tenfold serially diluted T4 plaquing (right). **h**, IP–MS/MS analysis of proteins that co-precipitate with N-terminally Flag-tagged CmdC at 0 and 15 min after infecting cells harbouring *cmdTAC* with T4 phage. **i**, Growth of uninfected *E. coli* cells harbouring *cmdTAC*, with expression of Gp23 and Gp31 as indicated. The centre line indicates mean, shaded area represents 95% confidence intervals. *n* = 3 biological replicates.

of *cmdTA*$_{3×HA}$ in a Δ*clpP* strain and monitored CmdA levels by immunoblot. Unlike in wild-type (MG1655) cells, CmdA accumulated to consistently high levels in the isogenic Δ*clpP* background (Fig. 2c). This accumulated CmdA was sufficient to neutralize CmdT, as reflected by the substantially improved plating viability of Δ*clpP* cells that express *cmdTA* compared with wild-type cells that express *cmdTA*.

We also found that co-expressing *cmdC*, either operonic with *cmdTA* (Fig. 2a) or separately from another plasmid (Fig. 2d), was sufficient to rescue the lethality of *cmdTA*, and immunoblotting confirmed that CmdC production promoted the accumulation of CmdA (Fig. 2d). Collectively, our findings indicate that CmdTAC is a TAC system with the following features: CmdT is a toxin that requires CmdA to fold and stably accumulate; CmdA serves as an antitoxin to neutralize CmdT in the presence of the SecB-like chaperone CmdC; and in the absence of CmdC, CmdA is degraded by the ClpP protease, leading to CmdT toxicity.

## CmdTAC defence requires CmdA degradation

Upon phage infection, CmdT must somehow be released from the CmdTAC complex to abort the infection and prevent the production of new

viruses. To understand the dynamics of the CmdTAC system, we used T4 to infect *cmdTAC* cells in which CmdT and CmdA encoded Flag and 3×HA epitope tags, respectively. We then monitored the abundance of each component by immunoblotting. CmdT was present throughout the course of phage infection, whereas CmdA levels steadily decreased (Fig. 2e and Extended Data Fig. 3d,e).

The decrease in CmdA during phage infection is likely to reflect its degradation by ClpXP and suggests that the ClpP protease is essential for CmdT activation and phage defence. To test this hypothesis, we challenged wild-type or Δ*clpP* cells harbouring *cmdTAC* with T4 phage. We found that T4 robustly infected Δ*clpP* cells harbouring either *cmdTAC* or an empty vector, in stark contrast to wild-type cells harbouring *cmdTAC*, which exhibited strong defence against T4 (Fig. 2f). The defect in defence seen with the Δ*clpP* strain was largely reversed by inducing expression of *clpP* in *trans*. As with Δ*clpP* cells, overproducing CmdC in cells harbouring *cmdTAC* led to the accumulation of CmdA and a loss of defence against T4 (Fig. 2g). Collectively, our results indicate that CmdTAC is likely to form a stable complex in the absence of phage, and infection then triggers the ClpP-dependent degradation of CmdA and release of CmdT.

## The T4 capsid protein triggers CmdTAC

To identify the phage-derived trigger that activates CmdTAC, we selected for T4 mutants that could escape CmdTAC defence. All had one of two independent mutations that extended the C terminus of the protein encoded by *alt.-3*, an uncharacterized gene with no predicted function. The mutations extended the Alt.-3 coding sequence (normally 96 amino acids) to one of two alternative stop codons, leading to either a 117 (Alt.-3†) or 110 (Alt.-3††) amino acid protein that overcame anti-phage defence (Extended Data Fig. 4a,b). Ectopically producing Alt.-3 was not sufficient to activate CmdTAC in the absence of phage infection, and a T4 strain lacking *alt.-3* did not escape CmdTAC defence (Extended Data Fig. 4c,d). Thus, we concluded that Alt.-3 is not an activator of CmdTAC and that the extended, mutant variants of Alt.-3 somehow inhibited CmdTAC activation, possibly by interfering with the proteolysis of CmdA. Indeed, we found that unlike wild-type Alt.-3, induction of Alt.-3† rescued CmdTA toxicity (Extended Data Fig. 4e) and promoted the accumulation of CmdA, similar to that observed during CmdC induction (compare Extended Data Fig. 4e to Fig. 2d). These results reinforce the conclusion that CmdT activation requires CmdA degradation and indicate that T4 phages can readily escape CmdTAC-mediated defence through mutations that inhibit this degradation.

To identify the bona fide activator of CmdTAC, we deleted *alt.-3* from the T4 genome and attempted to identify additional escape mutants by directed evolution, but were unable to. Because CmdC is needed to protect CmdA from degradation (Fig. 2d), we suspected that CmdC might preferentially bind a phage product instead of CmdA during infection. This would lead to release and degradation of CmdA, and the consequent liberation of CmdT. To identify potential CmdC interaction partners, we Flag-tagged CmdC in the context of *cmdTAC* under its native promoter and performed IP–MS/MS at 0 and 15 min post-T4 infection. As expected, before phage infection CmdC co-precipitated both CmdA and CmdT (Fig. 2h and Supplementary Table 1). However, at 15 min post-infection, the major co-precipitating protein was Gp23, the major capsid protein of T4[26].

To validate our IP–MS/MS result, we induced Gp23 expression from a high-copy plasmid in cells also containing *cmdTAC* in the absence of phage infection. The ectopic production of Gp23 alone leads to protein aggregation and cellular toxicity, as Gp23 requires an additional T4-encoded co-chaperonin, Gp31, to fold properly[27,28] (Fig. 2i). We therefore tested whether co-producing Gp23 and Gp31 was sufficient to activate CmdT. Indeed, co-producing Gp23 and Gp31 strongly inhibited the growth of *cmdTAC*-containing cells, but not an empty vector control (Fig. 2i). Of note, Gp31 alone does not result in CmdT-dependent growth inhibition. Collectively, these results strongly indicated that Gp23 activates the CmdTAC system.

## CmdT is an mRNA ADP-ribosyltransferase

As noted above, the CmdT toxin is likely to be an ADP-ribosyltransferase. To test for ADP-ribosylation, we first collected protein from cells producing CmdTA or from CmdTAC-containing cells infected with T4. We performed western blots with an ADP ribose-specific antibody but did not detect any CmdT-dependent protein bands (Fig. 3a). An electrostatic model of CmdT has a large positively charged cleft surrounding the putative active site (Fig. 3b), suggesting that CmdT might bind nucleic acids. To test for DNA modifications, we isolated DNA from the cells in the same conditions as the previous experiment and performed a dot blot with an ADP ribose antibody. However, we did not detect a signal, except from cells expressing the known DNA-targeting ADP-ribosyltransferase DarT[29] (Fig. 3a). We then repeated the procedure using total RNA, and observed strong signal on the anti-ADP ribose blot for cells harbouring *cmdTAC* and infected with T4 and for cells expressing *cmdTA* (Fig. 3a), but not in the empty vector controls, suggesting that CmdT was ADP-ribosylating RNA. Cells expressing *cmdT\*A*,

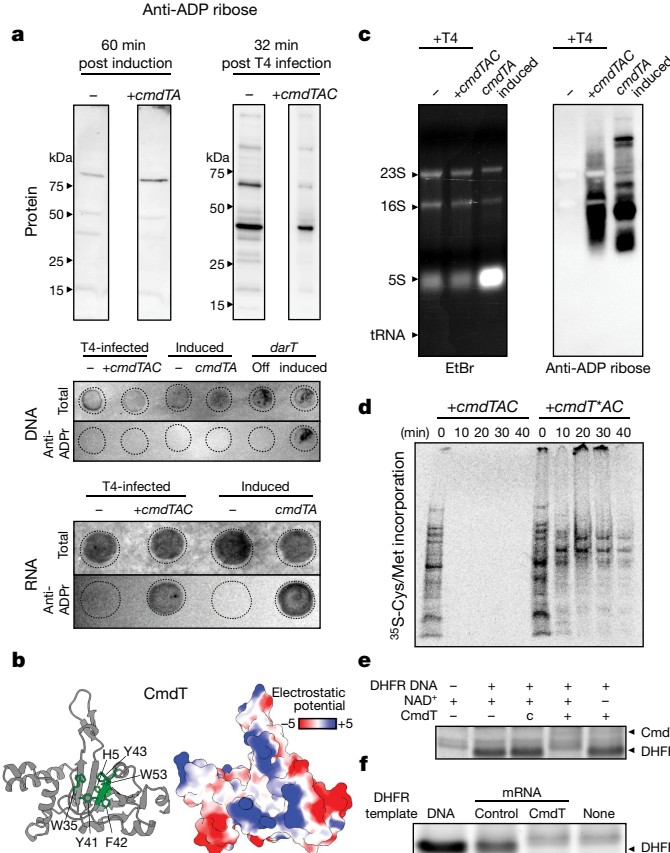

**Fig. 3 | CmdT is an ADP-ribosyltransferase that specifically targets mRNA to block translation. a**, ADP ribose antibody was used for immunoblots (top) or DNA and RNA dot blots (bottom) as indicated. Top left, samples taken from cells expressing *cmdTA* or harbouring an empty vector, 60 min post-induction. Top right, samples taken from cells expressing *cmdTAC* under its native promoter or harbouring an empty vector and infected with T4 phage for 32 min. Proteins were separated by SDS–PAGE; DNA and RNA samples were spotted onto nylon membranes. Total RNA and DNA are indicated by methylene blue staining. **b**, AlphaFold2-predicted structure of CmdT (left) with conserved aromatic residues and putative catalytic residues (see Extended Data Fig. 3a) shown in green, with the corresponding electrostatic surface representation. Colour bar is in units of kcal/(mol·e) (right). **c**, RNA samples from indicated strains and conditions were resolved on agarose gels and then stained with ethidium bromide (EtBr) to visualize total RNA or blotted with anti-ADP ribose to visualize ADP-ribosylated RNA. **d**, Protein synthesis, as measured by [35]S-labelled cysteine and methionine incorporation at the indicated timepoints during T4 infection of cells harbouring CmdTAC or CmdT\*AC, which has a catalytically inactive CmdT. Proteins were resolved by SDS–PAGE before phosphorimaging. Representative image from two independent biological replicates. **e**, In vitro transcription–translation reactions with DNA encoding dihydrofolate reductase (DHFR) used as template. Purified CmdT and NAD+ were added, where indicated, with DHFR production visualized by Coomassie staining. c indicates control eluant (mock purification). The DHFR band was identified based on the ladder in Supplementary Fig. 1. **f**, In vitro transcription–translation reactions using a DNA template encoding DHFR or mRNA templates produced by T7 RNA polymerase and treated with CmdT (or eluent from a mock purification of untagged CmdT) prior to addition. The DHFR band was identified based on positive and negative controls in lanes 1 and 4, respectively.

which produces CmdT with a putative active site mutation, lacked ADP-ribosyltransferase activity against RNA, as expected (Extended Data Fig. 5a).

To identify the RNA species that were being modified, we probed northern blots with the ADP ribose antibody (Fig. 3c). We detected strong signals in the regions corresponding to mRNA, but not those

corresponding to rRNA or tRNA, indicating that CmdT preferentially ADP-ribosylates mRNA. We then probed total RNA collected at multiple timepoints over the course of T4 infection and observed ADP-ribosylation of RNA within 10 min after infection (Extended Data Fig. 5b). We also tested whether inducing Gp23 and Gp31 led to ADP-ribosylation of RNA in CmdTAC-containing cells. Among all combinations of *cmdTAC*, *gp23* and *gp31*, we could detect RNA modification only when all three were expressed (Extended Data Fig. 5c). Collectively, our results indicate that CmdT is an RNA-targeting ADP-ribosyltransferase that is activated by the presence of T4 Gp23 and is dependent on Gp31.

## CmdT blocks translation and T4 replication

To understand how mRNA ADP-ribosylation inhibits phage development, we measured translation by growing T4-infected cells producing CmdTAC or CmdT*AC in the presence of radiolabelled cysteine and methionine. By 10 min post-infection we observed a complete block in incorporation in cells with CmdTAC but not CmdT*AC (Fig. 3d). To further test whether CmdT activity blocks translation, we purified CmdT–His$_6$ and, as a control, we isolated protein from a mock purification of CmdT lacking an affinity tag. Purified CmdT, but not the control eluant, blocked translation of a model protein, DHFR, in a cell-free in vitro transcription–translation reaction, and inhibition was dependent on NAD$^+$ (Fig. 3e). To ensure that CmdT had blocked DHFR translation and not transcription, we first in vitro transcribed DHFR and then treated the mRNA with purified CmdT and NAD$^+$ (Extended Data Fig. 6a). Providing this template to the in vitro translation reaction again resulted in no DHFR protein (Fig. 3f). By contrast, purified DHFR mRNA treated with the control eluant was robustly translated (Fig. 3f). Together, these data suggest that CmdT modifies mRNA to block translation.

To understand the global effect of CmdT activation on the T4 life cycle in vivo, we performed RNA sequencing (RNA-seq) on cells containing *cmdTAC* or an empty vector at 15 and 30 min post-infection with T4. Compared with the empty vector control, cells with *cmdTAC* showed substantially decreased expression of late genes and, consequently, increased read counts from early and middle genes (Extended Data Fig. 6b). The various stages of T4 gene expression depend on the successful translation of gene products in earlier stages of the life cycle[30] suggesting that T4 replicating in *cmdTAC*-positive cells is unable to progress to the late stages.

We also performed RNA immunoprecipitation with sequencing (RIP–seq) by enriching for ADP-ribosylated RNAs with an ADP ribose-specific antibody, followed by deep sequencing. Transcripts per million reads (TPM) values for T4 mRNAs were significantly higher in the immunoprecipitated samples compared with the control RNA-seq sample 30 min post-infection ($P = 0.04$; two-sided Welch's $t$-test). In addition, the TPM enrichment was significantly higher for T4 mRNAs than for tRNAs and rRNAs (Extended Data Fig. 6c). These findings reinforce our conclusion that CmdT is mRNA-specific. On the basis of our findings, we conclude that the ADP-ribosylation of mRNA by CmdT leads to a halt in translation, preventing the progression of T4 to the late stages of its life cycle and a failure to produce mature progeny.

## CmdT modifies GA dinucleotides in ssRNA

To determine the specificity of nucleic acid modification by CmdT we performed in vitro ADP-ribosylation using purified CmdT, NAD$^+$ and a 24-residue model mRNA oligonucleotide that contained a Shine-Dalgarno sequence and an AUG start codon. We found that CmdT treatment resulted in a shift in the migration of the single-stranded RNA (ssRNA) substrate on denaturing gels, indicative of covalent RNA modification (Fig. 4a). When a ssDNA oligonucleotide with the equivalent sequence was supplied as a substrate, a product was formed, but with substantially reduced kinetics (Fig. 4b). Supplying double-stranded RNA (dsRNA) substrates did not produce a higher mass product, even at high concentrations of CmdT (Fig. 4c).

To further test our hypothesis that mRNAs are preferentially targeted, we incubated purified CmdT with total RNA extracted from cells infected with T4, supplying 6-biotin-17-NAD$^+$ as a substrate. We then enriched for ADP-ribosylated RNA by precipitation with streptavidin beads and sequenced RNA samples taken before and after streptavidin pulldown, without rRNA depletion (Extended Data Fig. 6d). The results of this experiment paralleled those of the in vivo RIP–seq (Extended Data Fig. 6e), with enrichment of mRNAs compared with tRNAs and rRNAs. Together, these data suggested that unstructured, ssRNA is the preferred substrate of ADP-ribosylation by CmdT.

To determine the nucleotide and sequence specificity of CmdT, we repeated the ADP-ribosylation reactions with CmdT, NAD$^+$ and one of four ssRNA oligonucleotide substrates, each missing one base. Only the no-U and no-C oligonucleotides produced slower migrating products, suggesting that neither C nor U is required for modification (Fig. 4d). Notably, the no-U and no-C substrates produced four and three products, respectively, each of decreasing mobility in the gel (Fig. 4d and Extended Data Fig. 7a). This observation suggested that CmdT was adding multiple modifications to individual oligonucleotides. The formation of the slower migrating bands was dependent on reaction time (Extended Data Fig. 7a). We noticed that the number of bands formed was the same as the number of 5'-AG-3' or 5'-GA-3' dinucleotides in the no-U and no-C oligonucleotides (Fig. 4d). Similarly, the model RNA used in Fig. 4a–c contains one GA site and produced one reaction product.

To further test the sequence specificity of CmdT, we designed five ssRNA oligonucleotides with invariant C- and U-containing scaffolds. At two separate positions within the oligonucleotides, we introduced various dinucleotides: CC (control), AA, AG, GG or GA. During ADP-ribosylation by CmdT, substantial product was produced only with the substrate containing GA (Fig. 4e). Very faint bands appeared when the variable dinucleotide contained two purines other than GA, but none when the oligonucleotide contained two pyrimidines (CC). Taken together, our results indicate that CmdT preferentially recognizes GA dinucleotides.

To determine which nucleotides are covalently modified with ADP ribose, we performed a CmdT ADP-ribosylation reaction on an equimolar mixture of the no-U and no-C RNA oligonucleotides. We then digested the ADP-ribosylated RNA using nuclease P1, which hydrolyses 3'-5' phosphodiester bonds, and antarctic phosphatase, which removes 5' and 3' phosphates, and resolved the products with high-performance liquid chromatography (HPLC) (Extended Data Fig. 7b and Methods). In addition to peaks corresponding to C, U, A and G nucleosides, we observed other peaks with longer retention times (Fig. 4f, top row). These additional peaks were not seen in control reactions lacking NAD$^+$ or lacking CmdT. Notably, the absorbance of the adenosine peak dropped substantially in CmdT-treated samples compared with the control reactions, suggesting that the four new peaks were derived from modified adenosine (Extended Data Fig. 7c).

In parallel, we incubated the same reactions with snake venom phosphodiesterase I (SVPD), in addition to the nuclease and phosphatase. SVPD cleaves the phosphodiester bond in ADP ribose, in addition to some phosphodiester bonds that may be inaccessible to nuclease P1 (Extended Data Fig. 7b). For example, SVPD treatment of ADP-ribosylated adenosine would produce adenosine-ribose and adenosine, with the ribose and lone adenosine originating from SVPD cleavage of the ADP ribose moiety (Extended Data Fig. 7d). For the samples treated with CmdT and NAD$^+$, SVPD treatment produced two peaks with relative retention times corresponding to adenosine-ribose and guanosine-ribose[12], with the former showing stronger signal (Fig. 4f and Extended Data Fig. 7e). To identify the original (pre-SVPD treatment) peaks other than A, C, G and U nucleosides (Fig. 4f, top row), we isolated fractions corresponding to each, treated them with SVPD and then re-ran HPLC. This analysis indicated

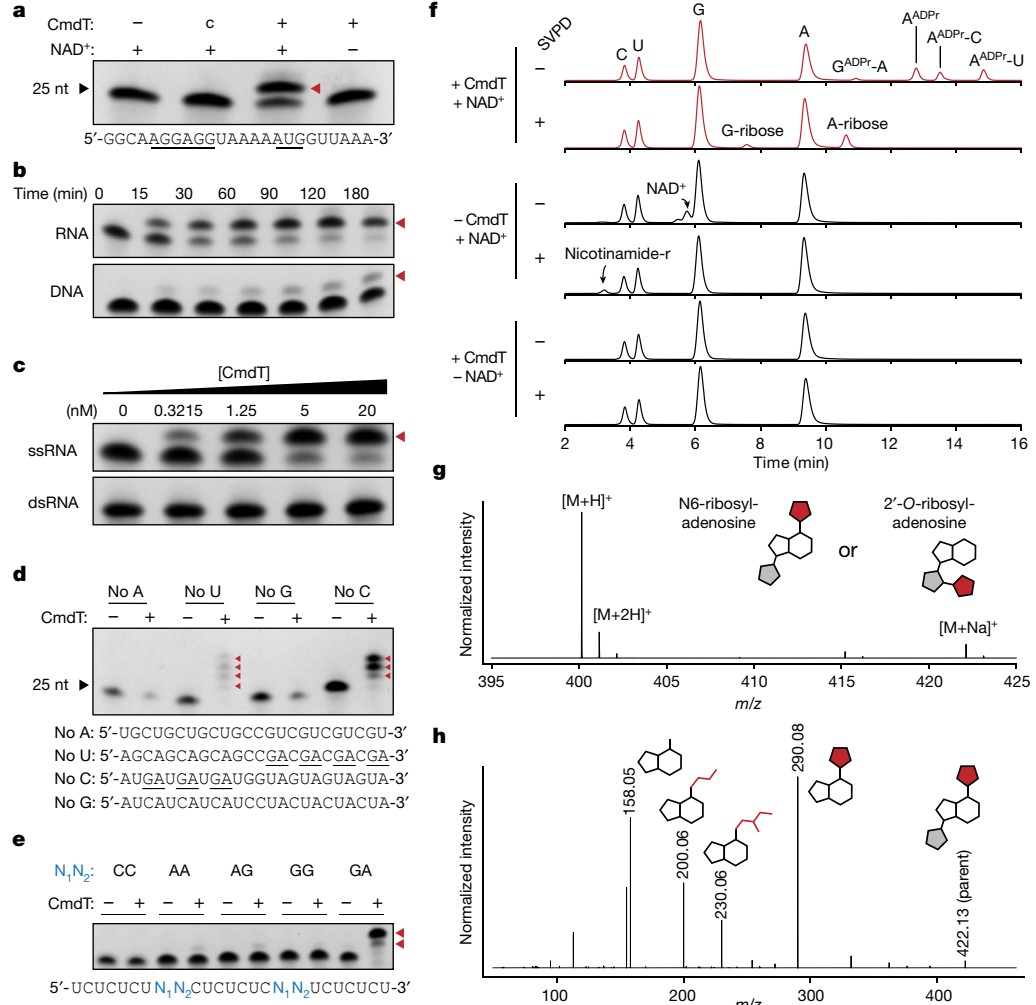

**Fig. 4 | CmdT modifies the N6 methyl group of adenine in GA dinucleotides of ssRNA. a**, The indicated RNA oligonucleotide was incubated with 4 nM CmdT–His$_6$, protein from a mock purification of untagged CmdT (c) or no protein (−), with or without NAD$^+$. Reaction products were resolved on a polyacrylamide TBE-urea gel and imaged by methylene blue staining. Red arrowheads indicate ADP-ribosylated products. **b**, ADP-ribosylation by 4 nM CmdT–His$_6$ of the RNA (top) or equivalent DNA (bottom) oligonucleotide at the indicated timepoints, visualized as in **a**. **c**, ADP-ribosylation by CmdT–His$_6$ of the ssRNA oligonucleotide from **b** or the corresponding dsRNA with increasing concentrations of CmdT–His$_6$ for 60 min. **d**, ADP-ribosylation by CmdT–His$_6$ of the indicated oligonucleotides, which lack A, U, G or C. **e**, ADP-ribosylation by CmdT–His$_6$ of the indicated oligonucleotide (bottom) with the identity of the variable dinucleotides (N$_1$N$_2$) indicated above each reaction. **f**, HPLC analysis of nucleosides isolated following incubation of the no-U and no-C RNA oligonucleotides in **d** with CmdT and NAD$^+$, as indicated, and then treated with nuclease P1 and antarctic phosphatase, with or without SVPD, as indicated. Peaks corresponding to A, C, G and U nucleosides, NAD$^+$, G-ribose, A-ribose and nicotinamide-riboside (nicotinamide-r, produced from SVPD cleavage of unused NAD$^+$) are marked, along with species from the top row identified in Extended Data Fig. 7e as GA, A, AC and AU, with the first nucleotide in each case being ADP-ribosylated (ADPr). The $y$ axis represents absorbance at 254 nm. **g**, ESI-MS analysis of the A-ribose peak fraction from **f**. **h**, MS/MS fragmentation of the A-ribose sodium adduct from **g**. Collision energy, 40 eV. Predicted fragments are annotated with structures and $m/z$ is indicated. Grey, ribose from adenosine; red, ribose from ADP ribose; white, adenine. Structures of fragmentation products are shown in Extended Data Fig. 8a.

that these other peaks were A, AC and AU, with the adenosine being ADP-ribosylated in each case, and a minor peak of GA with the G being ADP-ribosylated. Notably, the three major nuclease P1-resistant species each contained adenosine preceded by a G in the original substrate (Extended Data Fig. 7f), indicating that CmdT primarily ADP-ribosylates adenosine nucleotides at GA motifs in single-stranded mRNA.

To confirm the identity of the peak we inferred to be adenosine-ribose (Fig. 4f, second row), we performed electrospray ionization mass spectrometry (ESI-MS), which revealed peaks at 400 and 422 $m/z$, equivalent to the molecular weight of adenosine-ribose [M+H]$^+$ and [M+Na]$^+$ (Fig. 4g). There are two positions on adenosine that are most likely to react with NAD$^+$ and become ADP-ribosylated, the 2′ hydroxyl of the ribose and N6 on the adenine base (Fig. 4g). To distinguish between these, we analysed the adenosine-ribose sodium adduct by ESI-MS/MS

fragmentation (Fig. 4h and Extended Data Fig. 7g). We did not detect a fragment ion for ribose-2′-O-ribose ($m/z$ 287) which would be produced if the 2′-OH was ADP-ribosylated[12]. Instead, we observed peaks at 200 and 230 $m/z$, which are likely to correspond to fragments in which the adenine base contains a portion of the ribose derived from ADP-ribosylation of N6 (Extended Data Fig. 8a,b).

Finally, we compared the UV absorbance ($\lambda_{max}$) of adenosine and CmdT-generated adenosine-ribose. ADP-ribosylation at the 2′ hydroxyl of ribose does not shift the UV absorbance spectrum of adenosine[12], whereas modifications of the N6 position, such as N6-methylation or N6-isopentenylation, redshift the UV absorbance of adenosine by 6 or 8 nm, respectively[31]. The adenosine-ribose produced by CmdT shifted the $\lambda_{max}$ from approximately 258 nm to approximately 265 nm (Extended Data Fig. 8c). Given our mass spectrometry and UV spectroscopy data, along with the finding that CmdT can weakly modify ssDNA

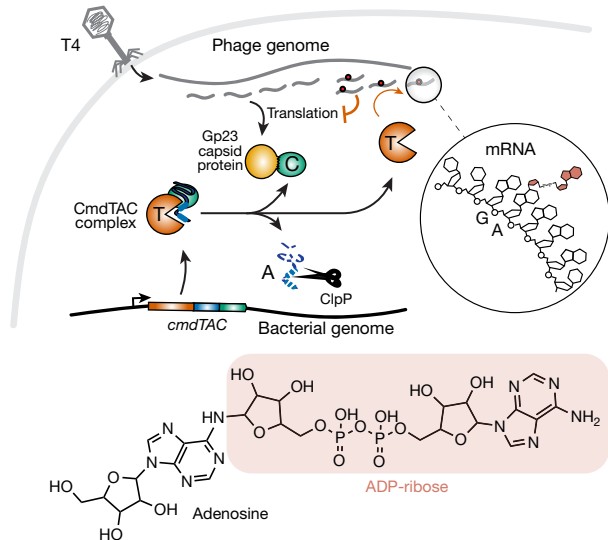

**Fig. 5 | Model for anti-phage defence by the CmdTAC system.** Top, CmdTAC produced by *E. coli* forms a complex prior to infection. Following T4 infection, newly synthesized capsid protein (Gp23) binds CmdC chaperone, leading to degradation of CmdA by ClpP and subsequent release of active CmdT toxin. CmdT ADP-ribosylates GA dinucleotides in ssRNAs thereby preventing their translation and the production of mature T4 virions. Bottom, structure of ADP-ribosylated adenosine.

(which lacks a 2′ hydroxyl), we conclude that the site of ADP-ribosylation by CmdT is the N6 of adenine in GA dinucleotides within ssRNA (Fig. 5).

## Discussion

This work reveals an ADP-ribosyltransferase that specifically targets mRNA. Prior studies have identified numerous ADP-ribosyltransferases in phages, bacteria and eukaryotes that use the highly abundant and reactive molecule NAD[+], or occasionally NAD-capped RNAs, to covalently modify a wide range of proteins, often to reversibly regulate their activities[5–7,32]. Protein ADP-ribosyltransferases are also featured in many biological conflicts with secreted toxins, such as pertussis and cholix toxins, that are capable of shutting down key cellular processes by modifying specific target proteins[33,34]. More recently, ADP-ribosyltransferases that target DNA and structured RNAs have been found[11–13]. Our findings, and other recent results[35], now extend the range of targets to include mRNA, with CmdT-catalysed ADP-ribosylation leading to a potent block in protein translation. Notably, in vitro studies have shown that human PARP10, PARP11, PARP14 and PARP15 can each modify ssRNA substrates[10,36], although there is no evidence for RNA modification occurring in vivo. Nevertheless, these prior results suggest that ADP-ribosylation of mRNA may be a widespread, but previously unappreciated modification occurring throughout biology. PARP10, PARP11 and PARP14 are encoded by interferon-stimulated genes[37–39] and several mammalian viruses, including SARS-CoV-2, encode ADP-ribosylhydrolases that offset poly-ADP-ribosyltransferase activity to promote viral replication[40,41]. Thus, the ADP-ribosylation of viral RNAs may be a broadly conserved and critical facet of anti-viral immunity.

Our results indicate that CmdT primarily modifies the N6 position of adenines within GA dinucleotides. Notably, GA nucleotides are almost always found within the Shine-Dalgarno sequences of bacterial (AGGAGG) and T4 (GAGG) transcripts. ADP-ribosylation of Shine-Dalgarno sequences may prevent modified transcripts from engaging ribosomes and initiating translation. Alternatively, or in addition, the modification of GA dinucleotides within an mRNA may disrupt base-pairing with tRNAs or may stall ribosomal translocation.

Further investigation is required to understand exactly how the block to translation occurs.

Many anti-phage defence systems have been discovered recently[3], most of which function through an Abi mechanism in which infected cells kill themselves or stop growing to prevent the production of new phages. Abi-based systems must remain off in the absence of infection and then be rapidly triggered upon phage infection. Our work reveals the T4 major capsid protein Gp23 as the probable direct activator of the CmdTAC system (Fig. 5). Structural proteins appear to be common triggers of anti-phage defence systems[42,43] possibly because they adopt folds that differ substantially from endogenous host proteins, thereby providing requisite specificity. Additionally, structural proteins accumulate to high levels, helping to ensure defence system activation. One potential downside to structural proteins as triggers is that most are encoded by late genes, raising the possibility that activation may not occur before new virions are produced. However, the 'late' classification is based on the peak time of expression, and both RNA-seq and mass spectrometry studies indicate that late genes are expressed at earlier timepoints[44,45]. Consistent with this idea, we detected CmdT activation and the ADP-ribosylation of mRNAs as early as 10 min post-infection. Precisely how the capsid protein interacts with CmdC and the specific regions involved will require further biochemical studies. Notably, despite extensive directed evolution we did not isolate any escape mutants of *gp23*, an essential gene. This could indicate that mutations to *gp23* that escape binding to CmdC are lethal by preventing the production of stable, mature virions. In addition, CmdC may engage an extended region of the capsid protein such that multiple substitutions and complex mutations would be required to abrogate binding.

CmdTAC adds to the growing list of toxin–antitoxin systems shown to function in anti-phage defence[14] and represents one of the first such TAC systems[35]. TAC systems have been well-characterized outside the context of phages[17–20], revealing principles such as the reliance of their antitoxins on C-terminal chaperone-addiction domains for folding and stability, and the use of SecB-like chaperones. The toxins of TAC systems show more variability, including several different families of endoribonucleases, which could, in principle, function similarly to CmdT to inhibit translation and prevent phage replication. More broadly, our discovery of ADP-ribosyltransferase activity in CmdT underscores the tremendous diversity in the mechanisms of action of toxins associated with toxin–antitoxin and TAC systems.

The intense conflict between phages and the bacteria they prey upon has led to the evolution of sophisticated anti-phage defence systems with a wide range of molecular innovations. Many of these systems have been identified in recent years, but the molecular mechanisms by which they inhibit phages remain incompletely defined. Our work reveals mRNA ADP-ribosylation as a potent defence mechanism used by bacteria to thwart phages. As several interferon-inducible mammalian PARPs—whose exact functions remain unclear—can also modify RNA, this mechanism may be broadly used in immunity throughout biology.

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

## Methods

### Statistics and reproducibility

Unless otherwise noted, representative images depict one of three biological replicates.

### Bacterial and phage growth and culture conditions

*E. coli* was routinely grown in LB medium at 37 °C unless otherwise stated. Phages were propagated and handled as described previously[4].

### Plasmid and strain construction

Primers, strains and plasmids are listed in Supplementary Tables 2, 3 and 4, respectively. In all cases, when plasmids were used as PCR templates, PCR samples were treated with DpnI at 37 °C for 1 h to eliminate the plasmid template before transformation. Finished DNA constructs were transformed into DH5α and verified with Sanger and/or long-read sequencing (Primordium) before transformation into the wild-type MG1655 background. For phage assays, the *cmdTAC* operon was present in low-copy pCD1 (Chloramphenicol-resistant (Cm$^r$), pSC101 origin of replication) and expressed from its native promoter (P$_{native}$). To construct the variant pCV49 (*cmdT\*AC*) by site-directed mutagenesis, complementary primers containing the Y to A mutation (CV109 and CV110) were used to amplify pCD2 for 15 cycles with KAPA DNA polymerase. pCV39 was constructed using the same primer set with pCD4 as the plasmid template.

Low-copy, P$_{native}$-*cmdTA$_{NT-3×HA}$C* (pCV43) was engineered using primers CV127 and CV128 to amplify pCD2 such that amplicon ends were located downstream of the CmdA start codon. A synthetic DNA fragment with three YPYDVPDYA codons plus GGGSGGG linker codons (3×HA tag, CV115) with ends complementary to the PCR-amplified vector was then ligated to this vector amplicon by Gibson assembly. P$_{native}$-*cmdTAC$_{NT-Flag}$* (pCV44) (NT Flag indicates an N-terminal Flag tag) was constructed by amplification of pCD2 with outward facing and 5′-phosphorylated primers CV120 and CV122 which included DYKD-DDDK codons followed by intramolecular blunt-end ligation with T4 DNA ligase. To construct P$_{native}$-*cmdT$_{CT-Flag}$AC*, primers CD41 and CD42 were used to amplify pCD2 and primers CD43 and CD44 were used to amplify C-terminally Flag-tagged (CT-Flag) CmdT sequence from pCD10 (see below for pCD10 construction) which was then ligated using Gibson assembly. To construct P$_{native}$-*cmdT$_{CT-Flag}$A$_{NT-3×HA}$C$_{NT-His6}$* (pCV42), first, a His$_6$ epitope tag encoding fragment was added to *cmdC* on pCD2 by the intramolecular blunt-end ligation strategy using primers CV120 and CV122. Next, primers CD41 and CD42 were used to amplify this plasmid and primers CD43 and CD44 were used to amplify CT-Flag *cmdT* from pCD10 which was then assembled using Gibson assembly. Finally, to insert three tandem HA tags onto the N terminus of CmdA, this intermediate construct was amplified by PCR with CV123 and CV124 such that amplicon ends were located downstream of the CmdA ATG start codon. The synthetic DNA fragment CV115 (HA tag) was used as a PCR template with primers CV125 and CV126 which was then ligated to the vector amplicon by Gibson assembly to produce pCV42.

To construct pCV45 used for the deletion of *alt.-3* from T4, complimentary oligonucleotides CV118 and CV119 with pCAS9 compatible overhanging sites were annealed by slow cooling from 98 °C in the presence of 50 mM NaCl to form a duplex spacer insert. pCAS9 and the annealed oligonucleotide were incubated with T4 DNA ligase and BsaI-v2 (NEB) in a one-pot reaction.

For pBAD30 constructs (kanamycin-resistant (Km$^r$), medium-copy p15a origin), primers CD5 and CD6 were used to amplify and linearize pBAD30. Insert fragments were amplified with the relevant primers (CD7-10, CD13-15, CD20-21, CD30-33, CD38-40) using T4 genomic DNA, plasmid DNA, or MG1655 genomic DNA as a template. pBAD-*cmdT$_{NT-HIS}$A* (pCD19) was created using PCR site-directed mutagenesis of pCD4 using primers CD45 and CD46. pBAD-*cmdT$_{NT-HIS}$* (Δ*cmdA*, pCD9) was created by using primers CV1 and CV2 (which exclude the open reading frame of *cmdA*) to amplify pCD19. This PCR amplicon was intramolecularly ligated with T4 DNA ligase. In some experiments, pAJM677 (Km$^r$, p15A origin), a variant of pBAD, was used to express CmdTA due to its higher expression after induction with arabinose and tighter repression (pCV41). To engineer pCV41, pAJ677 was amplified and linearized with primers CV113 and CV114. The insert containing *cmdTA* was amplified from pCD4 with primers CV116 and CV117. Plasmid and insert fragments were ligated by Gibson Assembly. To add the 3×HA tag to *cmdA* in this context, pCV41 was used as a PCR template with primers CV127 and CV128, and this amplicon was ligated to CV115 fragment by Gibson assembly. Anhydrotetracycline inducible (P$_{tet}$) pIF (carbenicillin-resistant (Cb$^r$), low-copy pSC101 origin) and pKVS45 (Cb$^r$, p15A origin) constructs were similarly constructed by PCR amplification of the vectors using primers CD24 and CD25 and inserts (CD26-29, CD36-37), followed by Gibson assembly.

To construct the gp23 expression plasmid pCD16, primers CD16 and CD17 or CD18 and CD19 were used to amplify the high-copy origin from pUC19 (pMB1* origin) and pBAD30 without its origin, respectively. These fragments were assembled using Gibson assembly to create a high-copy inducible vector. Subsequently, the backbone was amplified by PCR using primers CD5 and CD6 and *gp23* was amplified by PCR from T4 genomic DNA using primers CD34 and CD35. The two fragments were ligated using Gibson assembly.

### Plaque and phage assays

Overnight cultures of indicated strains were mixed 1:80 with melted LB with 0.5% agar and then overlaid on plates containing LB with 1.2% agar. For plaque assays done with induction of an arabinose-inducible promoter, base layer plates contained 0.2% w/w arabinose. A tenfold dilution series of the indicated phage was spotted onto plates and the plates grown at 30 °C overnight and plaque-forming units (PFU) were enumerated. $\log_{10}$(protection) (Fig. 1c) was measured as $-\log_{10}$ EOP, where EOP is the ratio PFU$_{experimental}$/PFU$_{control}$, where the subscript indicates the conditions. Unless otherwise noted, experiments were performed in biological triplicate and representative images are shown.

To measure survival of strains infected with T4, overnight cultures were diluted to OD$_{600}$ 0.1. Cultures were grown to OD$_{600}$ 0.3 and then adjusted to $-3 \times 10^7$ colony-forming units (CFUs) in a 1 ml volume in a 1.7 ml Eppendorf tube. Cells were infected with T4 at an MOI of 10 and incubated at 37 °C with rotation. At 0 and 18 min post-infection, cells were pelleted and washed twice with PBS to remove excess phages. One-hundred microlitres of tenfold dilutions were spread onto LB agar plates with chloramphenicol or kanamycin and CFUs were quantified. Survival was measured as CFU ml$^{-1}$ at 18 min post-infection divided by CFU ml$^{-1}$ at 0 min post-infection. To combat progeny phages in the empty vector strain inhibiting CFU formation, all samples were plated with $-10^7$ chloramphenicol-treated, chloramphenicol-sensitive companion plating cells (MG1655).

ECOI assays were conducted by diluting overnight cultures to OD$_{600}$ 0.1 in 20 ml LB. Cultures were grown until they reached OD$_{600}$ 0.3–0.4 at which point they were infected with T4 at an MOI of 0.1. After 20 min, 1 ml was pelleted and washed twice with PBS. One-hundred microlitres of tenfold dilutions were mixed with 50 μl of indicator strain and 3 ml LB 0.5% agar and overlayed onto LB plates. To control for unadsorbed phages, a Δ*ompC* strain (OmpC is the receptor for T4) was assayed in parallel. CFUs were enumerated and ECOI was calculated as PFU ml$^{-1}$ of the *cmdTAC*-containing strain divided by PFU ml$^{-1}$ of empty vector, after subtracting PFU ml$^{-1}$ of the Δ*ompC* control experiment from each value.

To determine burst size, cell cultures of empty vector and CmdTAC-containing strains were grown in LB + 20 μg ml$^{-1}$ chloramphenicol in a water bath at 37 °C until OD$_{600}$ measured 0.5. L-Tryptophan was then added to 20 μg ml$^{-1}$ to each culture to assist adsorption of T4. One hundred microlitres of a $10^7$ PFU ml$^{-1}$ T4 stock were added to 9.9 ml of each culture and incubated without shaking for 2 min to allow adsorption. Next, for each culture, 100 μl T4-infected culture

from this adsorption flask was added to 9.9 ml LB + 20 µg ml⁻¹ chloramphenicol (flask A). Flask A was again diluted 1:10 into flask B, and again 1:10 into flask C. Five hundred microlitres from flask A was added to 200 µl ice-cold chloroform and vortexed for 10 s. Viable PFUs from this chloroform-treated sample represent unadsorbed phage (adsorption control). Next, 100 µl from each flask A (time 0 sample) or the adsorption control was mixed with 3.5 ml LB 0.5% agar maintained at 50 °C to which was added 50 µl of an overnight culture of indicator strain. This mixture was vortexed briefly and overlayed onto LB + 20 µg ml⁻¹ chloramphenicol + 1.2% agar plates. All flasks were then left to incubate in a shaking water bath at 30 °C. After 60 min, 100 µl from flask C of the empty vector strain and flask A of the +*cmdTAC* strain were overlayed with indicator strain on agar plates. After overnight incubation at 37 °C, plaques were enumerated, and normalized to the adsorption control. Burst size was recorded as the number of plaques from each plate multiplied by their dilution factor, and then divided by the number of plaques at time 0.

## Growth curves

For measuring growth during T4 infection, overnight cultures of +*cmdTAC* and empty vector cells were back-diluted 1:200 in 96-well plates and infected with T4 at the indicated MOIs. Cultures were grown at 37 °C with orbital shaking on a plate reader (Biotek) for 6 h. For ectopic expression of Gp23 and Gp31 with CmdTAC, overnight cultures were back-diluted to $OD_{600}$ of 0.05 in M9L + 0.2% w/w glucose + 100 ng ml⁻¹ anhydrotetracycline (aTc) and grown for 3 h at 37 °C to pre-induce Gp31. Cultures were then pelleted and resuspended at an $OD_{600}$ of 0.05 in fresh M9L + 0.2% w/w glucose + 100 ng ml⁻¹ aTc or M9L + 0.2% w/w glucose + 100 ng ml⁻¹ aTc. Cultures were grown at 37 °C with orbital shaking on a plate reader for 12 h.

## RNA extraction following phage infection

Overnight cultures of +*cmdTAC* and empty vector cells were back-diluted and grown at 37 °C to $OD_{600}$ between 0.2 and 0.3 before being infected with T4 at a MOI of 10. RNA was extracted from cells at multiple timepoints post-infection as previously described[44]. In brief, 1 ml of cells was mixed with 1 ml of boiling lysis buffer (SDS 2%, 4 mM EDTA pH 8) and incubated at 100 °C for 5 min before flash freezing in liquid nitrogen. Two millilitres of acid-buffered phenol solution (pH 4.3, Sigma) heated to 67 °C was added to thawed samples, vortexed, and then incubated at 67 °C for 2 min. Samples were spun down at 20,000*g* for 10 min and hot phenol extraction repeated on the collected aqueous layer. A third extraction was then done using 2 ml of acid-buffered phenol-chloroform solution (Ambion). RNA from the final extraction was then precipitated at −20 °C for at least 1 h or at −80 °C overnight with 1× volume isopropanol, 1/10× volume 3 M sodium acetate (pH 5.5, Thermo Fisher), and 1/100× volume GlycoBlue. RNA was pelleted by centrifugation at 4 °C and 20,000*g* for 30 min. Pellets were washed twice with 800 ml of ice-cold 70% ethanol, air-dried, and resuspended in 90 µl RNAse-free $H_2O$ (Thermo Fisher).

To remove DNA, 10 µl of 10× Turbo DNase buffer (Ambion) and 2 µl of Turbo DNase I (Ambion) was added to each sample and incubated at 37 °C for 20 min. An additional 2 µl of Turbo DNase I was then added, and samples again incubated at 37 °C for 20 min. RNA was extracted from this digest by precipitation with 3× volume ethanol, 1/10× volume 3 M sodium acetate (pH 5.5), and 1/100× volume GlycoBlue. Pelleting and washing were performed the same as described above. RNA yield was verified using a NanoDrop spectrophotometer.

## RNA extraction from non-infected cells

Cells were grown until desired conditions and then 900 µl of culture was mixed with 100 µl of stop solution (5% acid phenol, 95% ethanol) and inverted to mix. Samples were then spun down at 13,000*g* for 30 s, the supernatant removed, and pellets flash frozen in liquid nitrogen. To each pellet, 400 µl of TRIzol Reagent (Invitrogen) heated to 65 °C was

added and mixed using a thermomixer for 10 min at 65 °C and 2,000 rpm before freezing at −80 °C for at least 10 min. Samples were thawed and then centrifuged at 20,000*g* for 5 min at 4 °C to pellet any debris and the TRIzol solution moved to a new tube. RNA was purified using the Direct-zol RNA Miniprep kit (Zymo Research) following manufacture's protocol including optional on-column DNAse treatment. RNA yield was verified using a NanoDrop spectrophotometer.

## Immuno-northern blotting

Novex 6% TBE-urea gels in 1× TBE buffer (Invitrogen) were pre-run at 180 V for at least 50 min prior to sample loading. Each RNA sample was mixed with equal volume of Novex 2× TBE-urea sample buffer (Invitrogen), heated at 90 °C for 10 min, and then placed on ice for 2–3 min just before loading. Gels were run at 180 V for 30–50 min depending on expected product length. Gels were removed from casing and incubated in 40 ml 1× TBE with added 4 µl of SYBR Gold stain (Thermo Fisher) for 10 min. Gels were imaged on a ChemiDoc MP imaging system (Bio-Rad) set for SYBR Gold imaging. RNA was transferred from the gel to a Hybond-N⁺ nylon membrane (Cytiva) via semi-dry transfer at 0.38 A for 90 min. After transfer, RNA was bound to the membrane by exposure to 120,000 µJ of UV radiation in a Stratalinker UV Crosslinker. Membranes were then incubated with shaking in 0.2% iBlock (Invitrogen) in 1× PBST for 10 min at room temperature or overnight at 4 °C. Primary antibody treatment was done with Poly/Mono-ADP Ribose rabbit antibody (Cell Signaling Technologies) diluted 1:1,000 in 0.2% iBlock + 1× PBST either for 2 h at room temperature or overnight at 4 °C with shaking. Following primary antibody treatment, membranes were washed 3 times for 10 min each with 1× PBST. For secondary antibody treatment, membranes were incubated for 1 h with shaking at room temperature with goat anti-rabbit IgG (H + L) secondary antibody, HRP (Invitrogen) diluted 1:1,000 in 0.2% iBlock + 1× PBST. Membranes were then again washed 3 times for 10 min each in 1× PBST. Signal was developed using SuperSignal West Femto maximum sensitivity substrate (Thermo Fisher) and imaged on a ChemiDoc MP imaging system set for chemiluminescence detection. Dot blots were conducted identically except 250 ng DNA or 1 µg RNA were spotted on membranes.

For agarose immuno-northern blots, 0.8 g of agarose was melted in 66.7 ml of $H_2O$ and allowed to cool to 65 °C. 8 ml of 0.2 M (10×) 3-(*N*-morpholino)propanesulfonic acid (MOPS) buffer, 5.4 ml of formaldehyde and 5 µl of 10 mg ml⁻¹ ethidium bromide were added to the agarose, and a 14×12 cm gel was cast and allowed to cool. 4 µg RNA were added to 17 µl sample buffer (2 µl 10× MOPS, 4 µl formaldehyde, 10 µl de-ionized formamide, and 1 µl ethidium bromide) and samples were denatured at 80 °C for 10 min then cooled on ice for 5 min. Prior to sample loading, the empty gel was run at 115 V for 5 min. 2 µl of loading dye (50% glycerol, bromophenol blue and xylene cyanol) were added to each RNA sample. Samples were then electrophoresed at 100 V for 80 min in 1× MOPS buffer. The gel was visualized before soaking in $H_2O$ for 10 min followed by a 20-minute equilibration in transfer buffer (3 M NaCl, 0.01 N NaOH). RNA was transferred onto Hybond-N+ nylon membrane by upward capillary transfer at room temperature for 75 min in transfer buffer. Immunoblotting was performed as described above. All immunoblotting experiments were performed in at least biological duplicate.

## RNA immunoprecipitation and sequencing

Cells were collected and RNA collected as described above for infected cells. rRNA was removed using a previously described ribosomal RNA subtraction method[46]. rRNA-depleted RNA was then fragmented using sonication. For each sample to be sonicated, 4 µg of RNA was added to 100 µl 1× TE buffer (Sigma) in a 1.5 ml TPX microtube (Diagenode) and incubated on ice for 15 min. Tubes were then placed in a Bioruptor 300 sonicator water bath chilled to 4 °C for 60 cycles of 30 s on, 30 s off at high power setting. Every ten cycles tubes were briefly spun down in a microcentrifuge to ensure all liquid stayed below the water line in the

sonicator. Each sample was then brought to a total volume of 200 µl with RNAse-free water and then precipitated at −20 °C for at least 1 h or at −80 °C overnight with 600 µl 100% ethanol, 20 µl of 3 M sodium acetate (pH 5.5), and 2 µl GlycoBlue. RNA was pelleted by centrifugation at 4 °C and 21,000$g$ for 30 min. Pellets were washed twice with 800 ml of ice-cold 70% ethanol, air-dried, and resuspended in 90 µl RNAse-free $H_2O$.

ADP ribose RNA immunoprecipitation was based on a methylated RNA immunoprecipitation sequencing (MeRIP-seq) protocol for low-input samples[47]. One-hundred microlitres of Dynabeads Protein G beads were washed 3 times in IP buffer (150 mM NaCl, 10 mM pH 7.5 Tris-HCl, 0.1% NP-40 substitute). Ten microlitres of Poly/Mono-ADP Ribose rabbit antibody (Cell Signaling Technologies) was added to washed beads resuspended in 500 µl IP buffer and then incubated overnight at 4 °C with end-to-end rotation. Following incubation, antibody conjugated beads were washed twice with IP buffer and then resuspended in 500 µl IP buffer with 20 µg fragmented, rRNA-depleted RNA and 5 µl Superase-In RNAse inhibitor and incubated overnight at 4 °C with end-to-end rotation. Samples were then washed twice with 1 ml IP buffer, twice with 1 ml low-salt wash (50 mM NaCl, 10 mM pH 7.5 Tris-HCl, 0.1% NP-40 substitute), and twice with 1 ml high-salt wash (500 mM NaCl, 10 mM pH 7.5 Tris-HCl, 0.1% NP-40 substitute). For each wash, beads were incubated in the wash solution for 10 min at 4 °C with end-to-end rotation. After the final wash, beads were incubated in 200 µl RLT buffer from the Qiagen RNeasy kit for 2 min at room temperature with end-to-end rotation. Supernatant was separated from the beads using a magnetic rack, transferred to a new tube, and mixed with 200 µl of 100% ethanol. This mixture was passed through a RNeasy MiniElute spin column by centrifugation at 20,000$g$ at 4 °C for 1 min. Spin columns were then washed once with 500 µl RNeasy RPE buffer and once with 500 µl 80% ethanol with each spin done at 20,000$g$ for 1 min at 4 °C. Columns were then spun at 20,000$g$ for 5 min to remove residual ethanol. RNA was eluted from the column in 15 µl RNAse-free $H_2O$ with a spin at 20,000$g$ for 5 min at 4 °C. RNA yield and integrity was verified using a NanoDrop spectrophotometer and a Novex 6% TBE-urea gel (Invitrogen), respectively.

Pre- and post-immunoprecipitation RNA (50–100 ng) was then used to make RNA-seq libraries using the NEBNext Ultra II RNA Library Prep Kit for Illumina following the manufacturer's protocol for use with rRNA-depleted formalin-fixed, paraffin-embedded RNA. Paired-end sequencing of the libraries was performed on a Singular G4 machine at the MIT BioMicroCenter. FASTQ files were then mapped to the MG1655 genome (NC_00913.2), the T4 genome (NC_000866), and the plasmid pKVS45-CmdTAC as previously described[44,48].

## Library preparation for RNA-seq

Cells were collected and RNA collected as described above for infected cells. rRNA was removed using a previously described ribosomal RNA subtraction method[46]. One-hundred nanograms of each rRNA-depleted RNA sample was then used to make RNA-seq libraries using the NEBNext Ultra II RNA Library Prep Kit for Illumina following the manufacturer's protocol for use with purified mRNA or rRNA-depleted RNA. Paired-end sequencing of the libraries was performed on an Illumina NextSeq 5000 machine at the MIT BioMicroCenter. FASTQ files were then mapped to the MG1655 genome (NC_00913.2), the T4 genome (NC_000866), and the plasmid pKVS45-CmdTAC, as previously described[44,48].

## Co-immunoprecipitation and LC–MS/MS

Overnight cultures of +*cmdTAC* and +*cmdTA/Flag-C* or +*cmdT-Flag/AC* cells were back-diluted in 250 ml LB and grown at 37 °C to an $OD_{600}$ of 0.3 and then for +*cmdTAC and +cmdTA/Flag-C* samples infected with T4 at an MOI of 10. At 0 min for all samples and 15 min post-infection for +*cmdTAC* and +*cmdTA/Flag-C* cultures 64 ml of sample was pelleted by centrifugation at 7,500$g$ for 5 min. Pellets were decanted and resuspended in 1 ml of lysis buffer (25 mM Tris-HCL, 150 mM NaCl,

1 mM EDTA, 5% glycerol, 1% Triton X-100) supplemented with 1 µl ml$^{-1}$ Ready-Lyse Lysozyme (Fischer Scientific), 1 µl ml$^{-1}$ benzonase (Sigma), and cOmplete Protease Inhibitor Cocktail (Roche) and then flash frozen in liquid nitrogen. Samples were kept in liquid nitrogen until all timepoints were collected. Samples were subjected to two freeze-thaw cycles in liquid nitrogen to ensure complete lysis of cells. Additional lysis buffer was added to samples as needed to normalize sample concentrations by $OD_{600}$. Samples were spun at 20,000$g$ for 10 min at 4 °C to pellet any debris. For each sample, 50 µl of Pierce Anti-DYKDDDDK magnetic agarose beads was mixed with 450 µl lysis buffer and then collected to the side of the tube using a magnetic rack. Beads were then washed twice with 500 µl lysis buffer. After the final wash, beads were mixed with 1 ml of sample and incubated for 20 min at room temperature on an end-to-end rotor. After incubation, beads were washed in wash buffer (1× PBS, 150 mM NaCl) twice and then once with MilliQ $H_2O$.

On-bead reduction, trypsin digest, and LC–MS/MS were done by the MIT Biopolymers and Proteomics Core as previously described[42]. In brief, proteins were reduced for 1 h at 56 °C with 10 mM dithiothreitol (Sigma) and then alkylated for 1 h at 25 °C in the dark with 20 mM iodoacetamide (Sigma). Proteins were digested with modified trypsin (Promega) overnight in 100 mM, pH 8 ammonium bicarbonate at a 1:50 enzyme:substrate ratio. Formic acid (99.9%, Sigma) was added to stop trypsin digest. Digested peptides were desalted using Pierce Peptide Desalting Spin Columns (Thermo) then lyophilized. Peptides were separated on a PepMap RSLC C18 column (Thermo) over 90 min by reverse phase HPLC (Thermo Ultimate 3000) before nano-electrospray with an Orbitrap Exploris 480 mass spectrometer (Thermo). Mass spectrometer run was done in data-dependent mode. Full scan parameters were resolution of 120,000 across 375–1600 $m/z$ and maximum IT 25 ms. This was followed by MS/MS for as many precursor ions in a two second cycle with a resolution of 30,000, dynamic exclusion of 20 s, and a NCE of 28. Detected peptides were mapped to MG1655, plasmid, and T4 protein sequences and the abundance of proteins were estimated by number of spectrum counts/molecular mass to normalize for protein sizes. The ratio of spectral counts between the Flag pulldowns and untagged pulldowns at each timepoint were used to generate the data in the figures with a pseudocount added to each count.

## In vitro transcription and translation

In vitro transcription–translation assays were conducted using the PURExpress kit (NEB) according to the manufacturer's protocol with a 2-h incubation at 37 °C. Each reaction was supplemented with 1 U µl$^{-1}$ Riboguard RNase inhibitor (LGC Biosearch Technologies), with or without 1 mM NAD$^+$ and protein eluants as indicated. When supplying mRNA as a translation template, primers were used to amplify the DHFR gene using PCR from the PURExpress control DHFR plasmid. The PCR amplicon was purified using the DNA Clean & Concentrator Kit (Zymogen). Then, mRNA was synthesized from the PCR template by incubating 300 ng DNA with 200 U T7 RNA polymerase, 0.5 mM NTPs, and 5 mM DTT in a final reaction volume of 40 µl at 37 °C for 4.5 h. The resulting RNA was purified from the reaction using the RNA Clean and Concentrator Kit (Zymogen) with on-column DNase I treatment. Pure mRNA was then treated with CmdT or control mock purified protein in 1× ADPr buffer (20 mM Tris-HCl pH 8.0 and 150 mM NaCl) with 1 mM NAD$^+$ and 1 U µl$^{-1}$ Riboguard at 37 °C for 2 h. RNA was again purified as before, and 1 µg was supplied in the PURExpress reaction for 4 h at 37 °C. From this reaction, 2.5 µl was then denatured in Laemmli buffer and run on a 8–16% polyacrylamide gel by SDS–PAGE and stained with either Brilliant Blue R250 or Coomassie Fluor Orange (Molecular Probes) and visualized on a Bio-Rad ChemiDoc MP imager.

## Protein immunoblotting

Cell cultures were grown overnight and diluted 1:200 in fresh LB containing the appropriate antibiotics. Cultures were grown at 37 °C to mid-exponential phase and then treated with T4 at an MOI of 10, or

the appropriate inducers as dictated by the experiment. At various timepoints, cells were pelleted, flash frozen and subsequently resuspended in Laemmli buffer with 2.5% 2-mercaptoethanol in a volume normalized to culture turbidity (100 μl OD600$^{-1}$ ml$^{-1}$). Samples were run by standard SDS–PAGE on 12% polyacrylamide gels. Transfer onto 0.2 μm PVDF membranes was done at 90 V for 40 min for CmdA-HA, and otherwise was done at 100 V for 60 min. Membranes were blocked in Tris-buffered saline with 0.05% Tween-20 (TBST) and 5% non-fat milk for 60 min at room temperature and incubated with primary monoclonal antibody (1:1,000 rabbit anti-Flag or anti-HA, Cell Signaling Technologies) overnight at 4 °C. Membranes were washed with TBST and incubated with HRP-conjugated goat anti-rabbit IgG (Invitrogen) in blocking buffer for 60 min at room temperature. Membranes were again washed and incubated with SuperSignal West Femto Maximum Sensitivity Substrate (Thermo Scientific) before exposure on a Bio-Rad ChemiDoc MP imager. Membranes were stained with Brilliant blue R250 as a loading control. To quantify band intensity, we used the gel analysis tool in ImageJ. Pixel intensity from the antibody signal was normalized to pixel intensity of total protein stain.

For immunoblots, membrane chemiluminescence was imaged directly followed by imaging of pre-stained molecular weight markers. Images were aligned, as shown in Supplementary Fig. 1, to relate chemiluminescent bands to molecular weight markers, as shown in main figures.

Non-denaturing blots were performed by lysing cells with a buffer composed of 50% BPER-II (Thermo Scientific), 0.1 mg ml$^{-1}$ lysozyme, cOmplete protease inhibitor (Sigma-Aldrich), 6 U DNase I (NEB) and 3 μl RNase A (NEB) in volumes normalized to OD$_{600}$ of culture samples. Samples were incubated until clear at room temperature and then spun at max speed in a table-top centrifuge for 5 min to pellet insoluble material. Native loading dye (6×; 600 mM Tris-HCl, 50% glycerol, 0.02% bromophenol blue) was added to samples and loaded onto a 12% polyacrylamide Mini-Protean TGX pre-cast gel (Bio-Rad, does not contain SDS). Samples were electrophoresed in 25 mM Tris, 192 mM glycine running buffer at 75 V for 90 min. Transfer was conducted onto 0.2 μm PVDF membranes as described in this section at 100 V for 1 h at 4 °C. Blots were processed as described above. Blots shown are representative images of at least two biological replicates.

### T4 genome engineering and evolution
The evolution of T4 on *cmdTAC*-containing cells was performed as described previously[49] for four rounds, resulting in the *alt.-3* C-terminal extensions in all five replicates. To generate *alt.-3* mutants for further evolution experiments, T4 stock was overlayed onto strains containing Cas9 and spacers directed toward *alt.-3* or a control plasmid with no spacer (ML4233 and ML4234). The number of plaques formed on the spacer-containing strain was compared to the control to determine whether there was any selection imposed by the spacer. Despite attempting eight potential spacers, no selection was observed. To mitigate this, we repeated the experiment with T4 Δ*agt* Δ*bgt* (DNA contains non-glucosylated, 5-hydroxymethyl cytosine) on an *E. coli* Δ*mcrA* Δ*mcrBC* background required for viability of T4 Δ*agt* Δ*bgt*. This T4 formed fewer plaques in the presence of the *alt.-3* spacer, suggesting that selection for *alt.-3* mutants was imposed in this condition. The *alt.-3* region was amplified by PCR and Sanger sequenced from plaques that were able to form on the spacer-containing strain. Of those plaques, we isolated a strain that harboured a mutation encompassing nearly the entire open reading frame of *alt.-3*. The T4 Δ*alt.-3* strain was propagated in the presence of the spacer and stored as a stock at 4 °C. Evolution of this T4 strain on *cmdTAC*-containing cells was conducted the same as before, for 17 rounds, without observing mutations that increased plaquing ability.

### Radiolabel incorporation assays
Overnight cultures of +*cmdTAC* and +*cmdT\*AC* cells were back-diluted in LB + 20 μg ml$^{-1}$ chloramphenicol and grown at 37 °C to an OD$_{600}$ between 0.2 and 0.3. An aliquot of each culture was collected before T4 infection at an MOI of 10 and again at each indicated timepoint ($t$ = 10, 20, 30, 40 min post-infection). Each collected sample was incubated with EasyTag EXPRESS-$^{35}$S protein labelling mix (Perkin Elmer) at 23 μCi ml$^{-1}$ for 2 min at 37 °C. Labelling was chased with an unlabelled cysteine/methionine mixture at 5 mM and then samples precipitated in 13% w/v ice-cold TCA. Samples were pelleted by centrifugation at 16,000$g$ for 10 min at 4 °C, washed twice with 500 μl ice-cold acetone, and then resuspended in resuspension buffer (100 mM Tris pH 11.0, 3% w/v SDS). Samples were run on 4–20% SDS–PAGE gels (Bio-Rad), the gels incubated in Gel-Dry Drying Solution (Invitrogen) for 10 min, and then dried on a vacuum gel dryer for 2 h at 80 °C. Dried gels were exposed to a phosphorimaging screen overnight before imaging on an Amersham Typhoon imager.

### Protein purification
Five millilitres of cultures of ML4207 and ML4232 were grown overnight at 37 °C in LB + 0.2% glucose. The following day, 5 ml of each culture was washed of glucose twice and used to inoculate 495 ml of LB + 25 μg ml$^{-1}$ kanamycin. After 1 h of additional growth, arabinose was added to a final concentration of 0.2%. Cultures were grown an additional 95 min, pelleted, washed with H$_2$O, again pelleted, and flash frozen in liquid N$_2$. The following day, cell pellets were resuspended in 4 ml lysis buffer (50 mM Tris pH 7.5, 500 mM NaCl, 0.05% Tween-20, EDTA-free protease inhibitor, 0.5 mM PMSF, 0.5 mg ml$^{-1}$ lysozyme, 5 mM imidazole, and 5% glycerol) on ice. Cells were then lysed by sonication in a Bioruptor 300 for two rounds of 10 cycles each, high setting, 30 s on/30 s off. One millilitre of Ni-NTA agarose resin (Qiagen, 0.5 ml bed volume) was equilibrated in lysis buffer. Cell lysate was clarified by centrifugation then incubated with the Ni resin for 1 h at 4 °C with gentle rocking. The following steps were conducted at 4 °C. The resin was then passed through a 10 ml chromatography column and then washed 5× with 2.5 ml of wash buffer (same as lysis buffer but without lysozyme, and imidazole at 20 mM). Protein was then eluted 5× with 2.5 ml elution buffer (wash buffer with imidazole concentration at 300 mM). Eluted proteins were buffer-exchanged into Tris pH 7.4 using Micro Bio-Spin chromatography columns (Bio-Rad) and concentrated using Amicon Ultra 0.5 ml centrifugal filters with a 3-kDa cutoff.

### In vitro ADP-ribosylation by CmdT
A typical reaction was assembled on ice as follows. To a buffer composed of 20 mM Tris-HCl pH 8.0 and 150 mM NaCl, we added 1 U μl$^{-1}$ Riboguard RNase inhibitor, 1 mM NAD$^{+}$ (NEB), 4 μg of DNA or RNA oligonucleotide, and protein at the concentrations indicated. The reactions were then incubated in a thermocycler at 37 °C. To stop the reaction, an equal volume of 2× 6 M urea sample buffer (Novex) was added. RNA was denatured at 95 °C for 10 min and then immediately placed on ice. One microgram of RNA samples were then subject to electrophoresis in 15% polyacrylamide TBE-urea gels at 180 V for 75 min. Gels were stained both with SYBR Gold and with a concentrated solution of 0.2% methylene blue in 0.1× TBE buffer for 15 min, de-stained with several changes of H$_2$O and imaged.

### In vitro ADP ribose RNA pulldown
Twenty micrograms of total RNA were ADP-ribosylated with CmdT as described above with 0.25 mM 6-Biotin-17-NAD$^{+}$ (Cayman Chemical) for 4 h at 37 °C and then continued at 4 °C overnight. Two control reactions were set up identically except with mock purified protein, or with standard NAD$^{+}$ in place of 6-Biotin-17-NAD$^{+}$. Ten micrograms of each reaction were kept at −80 °C as the pre-pulldown sample. The remaining 10 μg were incubated with streptavidin conjugated superparamagnetic beads (Dynabeads MyOne Streptavidin C1) following the manufacturer's protocol. RNA was stripped from the beads by addition of 0.5 ml Trizol and incubation at 25 °C for 15 min on a thermomixer at 1,000 rpm. Beads were then precipitated with a magnet and 100 μl of chloroform

were added. The phases were separated by centrifugation at 14,000*g* for 15 min. Finally, the aqueous phase was purified using the RNA Clean and Concentrate Kit (Zymogen). Pre- and post-pulldown samples were electrophoresed on a 6% TBE–urea gel for 45 min, stained with SYBR Gold and imaged. The samples from pre- and post-pulldown reactions containing 6-Biotin-17-NAD[+] and CmdT were subject to RNA-seq as described in this section, but without rRNA depletion.

## HPLC analysis of ribonucleosides

Ten micrograms each of no-U and no-C RNA oligonucleotides (Fig. 4d) were subjected to ADP-ribosylation as described above. Controls were included in which purified CmdT was replaced by a mock purification, or in which NAD[+] was omitted. Next, samples were split and treated with either 100 U Nuclease P1 (NEB) and 10 U antarctic phosphatase, or, the same with the addition of 1 U Phosphodiesterase I from *Crotalus adamanteus* venom (SVPD, Millipore Sigma). Reactions were incubated in digest buffer (25 mM Tris-HCl pH 7.6, 50 mM NaCl, 1 mM ZnCl$_2$, and 10 mM MgCl$_2$) at 37 °C for 3.5 h in a total volume of 110 µl. One-hundred microlitres of digested and dephosphorylated nucleosides (10 µg) were injected onto a Vydac C18 4.6 ×250 mm reverse phase silica column (218TP54) equilibrated with 90% buffer A (0.1 M triethylammonium acetate (TEAA), pH 7.0)/10% buffer B (0.1 M TEAA, 20% acetonitrile, pH 7.0). HPLC was run with a mobile phase gradient composed of buffer A and B, 10–60% B from 1–21 min and 60–97% B from 21–26 min at a flow rate of 1 ml min$^{-1}$. Analytes were measured at A$_{254}$. On a replicate run, samples without SVPD treatment were collected as fractions and relevant fractions were lyophilized. The samples were then resuspended in digest buffer, and again incubated for 3.5 h with 10 U antarctic phosphatase and 1 U SVPD and analysed by HPLC as described above.

## Mass spectrometry of modified nucleosides

Fractions collected from HPLC analysis were dried in a speed-vac and resuspended in 200 µl of 50% acetonitrile in 0.1% formic acid. The fractions, or a buffer blank, were directly infused by syringe pump into a Thermo Q Exactive with an API source and electrospray ionization probe at a flow rate of 5 µl min$^{-1}$. The instrument was operated in positive ion mode. MS/MS was conducted at collision energies of both 25 and 40 CE. Instrument parameters were as follows: spray voltage, 3.8 kV; capillary tube temperature, 280 °C; sheath gas, 20; auxiliary gas, 5; sweep gas, 5.

## Bioinformatic analyses

The CmdT sequence logo was generated from the CmdT hmm file from DefenseFinder[50] using skylign.org.

Hidden Markov model profiles of CmdT and CmdC were downloaded from DefenseFinder[50] and searched against the RefSeq non-redundant protein database using hmmscan and default parameters. Protein hits were then identified in all available RefSeq bacterial genomes and CmdTAC was called if both CmdT and CmdC were present within two proteins of each other in the genome. CmdA was not included in calling as it both has higher sequence variability and is often unannotated in clearly homologous systems. All datasets were downloaded in July 2023. The complete taxonomic lineage of RefSeq genomes was created and filtered to include bacteria of current interest (genera with greater than 1,000 sequenced genomes). A taxonomic relationship of these genera was produced with NCBI Common Tree, and presence/absence was recorded from the taxonomic profiles of the CmdT/C hmmscan.

## Structural predictions

Protein homology was assessed using HHpred[51]. Predictions of the structures of individual components and CmdTAC as a complex were done using AlphaFold2[52] with the multimer module and default parameters on the reduced database with 1 prediction generated per model. Structural homology searches based on the AlphaFold2-predicted structures of CmdT and CmdC were done using Foldseek[53] and the

top hit for each search used for subsequent analyses. Electrostatics modelling was done using the coulombic function in ChimeraX. All predicted structure visualization was done using ChimeraX.

## RNA-seq and RIP-seq read mapping

FASTQ files for each sample were trimmed using cutadapt[54] (version 1.15) and then mapped to the *E. coli* MG1655 genome (NC_00913.2), the T4 genome (NC_000866), and the plasmid pKVS45-CmdTAC using bowtie2[55] (version 2.3.4.1) with the following arguments: -D 20, -I 40, -X 300, -R 3, -N 0, -L 20, -i S,1,0.50. Sam files generated from bowtie2 mapping were then converted to bam files using samtools[56] (version 1.7) and then further converted to numpy arrays using the genomearray3 python library[57] for use in downstream analyses. For in vivo RIP–seq analyses only highly expressed transcripts as determined by transcripts with an RNA TPM for both replicates greater than or equal to the minimum mean TPM of any T4 transcript were used. For statistical comparisons of TPM ratios between RNA types a Welch's *t*-test was used.

## Reporting summary

Further information on research design is available in the Nature Portfolio Reporting Summary linked to this article.

## Data availability

Summary spectra and raw data for IP–MS/MS of CmdC and CmdT pulldowns were deposited at MassIVE and can be accessed under accession MSV000093692; a summary table of high confidence hits is provided as Supplementary Table 1. Raw data for nucleotide MS and ESI-MS/MS were deposited at MassIVE and can be accessed under accession MSV000093878. RNA-seq and RIP–seq data are available at GEO under accession number GSE253514. All other data are available in the manuscript or the supplementary materials. Source data are provided with this paper.

## Code availability

Code used for RNA-seq and in vivo RIP–seq analysis is available at https://doi.org/10.5281/zenodo.10522487 (ref. 58).

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

**Acknowledgements** The authors thank D. Saxton, T. Zhang, S. Srikant and C. Beck for comments on the manuscript; B. Imperiali for help with interpreting the mass spectrometry data; C. Eickmann for help with AlphaFold2 predictions; the MIT BioMicroCenter and its staff for their support with sequencing; the MIT Biopolymers and Proteomics Core and its staff for assisting in HPLC and mass spectrometry experiments; and S. Srikant for help with T4 reverse

genetics. M.T.L. is an Investigator of the Howard Hughes Medical Institute. This work was also supported by a Howard Hughes Medical Institute Gilliam Fellowship awarded to C.R.D. and a National Institutes of Health NIGMS grant 5F32 GM139231-02 to C.N.V.

**Author contributions** C.R.D. and C.N.V. cultured bacteria and phages, created plasmids and strains, and conducted bacterial growth and phage plating assays. C.R.D. and C.N.V. performed RNA extractions and immuno-northern blots. RNA-seq, RIP–seq and all RNA-seq library preparation was performed by C.R.D. In vitro ADPr RNA-seq was performed by C.N.V. IP–MS was performed by C.R.D. In vitro transcription and translation, protein immunoblotting, and T4 genome engineering and evolution were performed by C.N.V. Radiolabel incorporation assays were performed by C.R.D. Protein purification, in vitro ADPr assays, HPLC and ESI-MS were conducted by C.N.V. Bioinformatic analyses were performed by C.R.D. and C.N.V.

Structural predictions were performed by C.R.D. C.R.D., C.N.V. and M.T.L. designed experiments, analysed data, prepared figures and wrote the manuscript.

**Competing interests** The authors declare no competing interests.

**Additional information**
**Correspondence and requests for materials** should be addressed to Michael T. Laub.

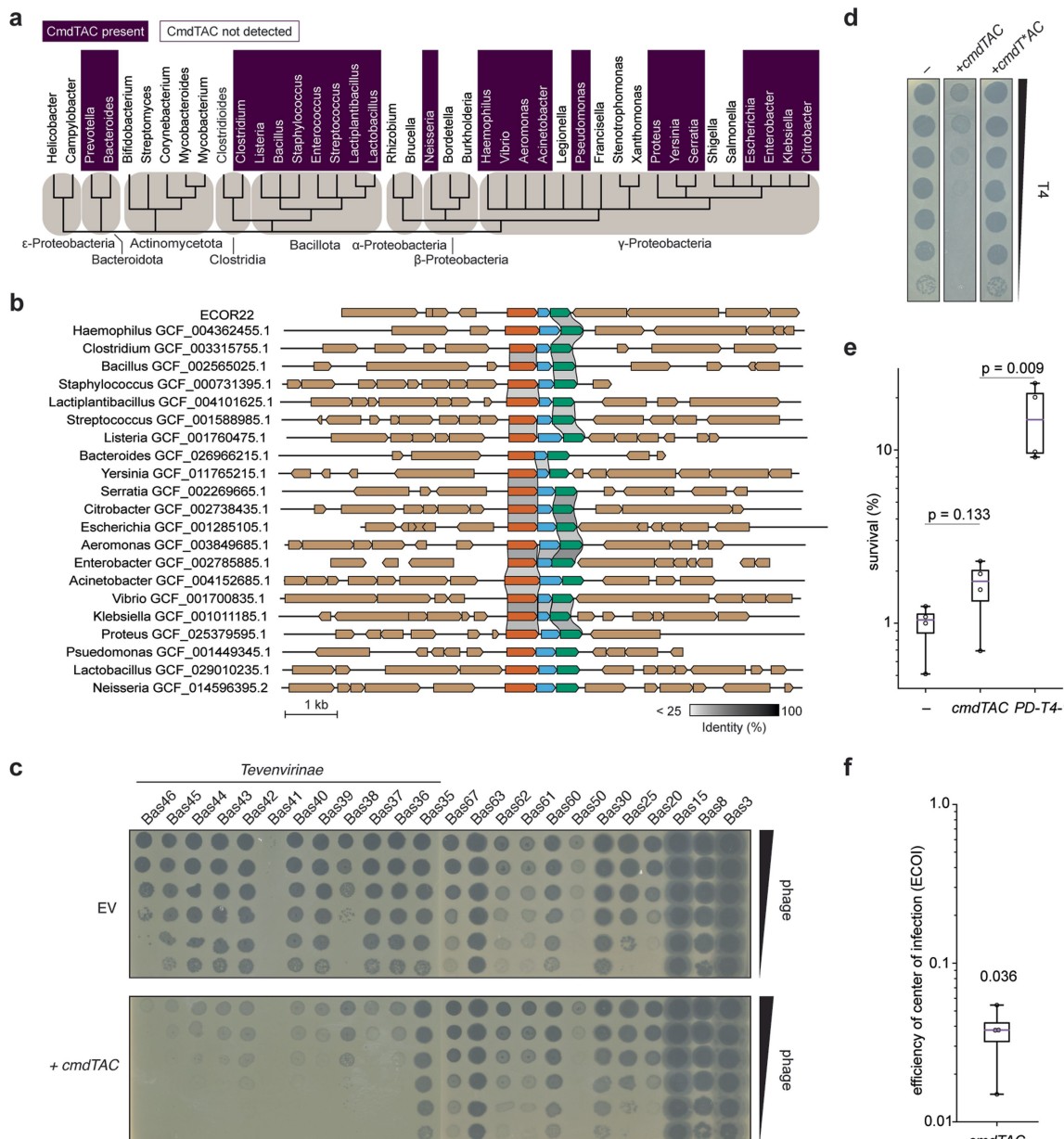

**Extended Data Fig. 1 | Taxonomic distribution of *cmdTAC* and efficiency of plaquing for BASEL and T-even phages.** (a) Presence or absence of CmdTAC homologs in bacterial genera with > 1000 sequenced genomes (see Methods). (b) Examples of CmdTAC homologs found in diverse bacterial species. Grey bars between genes capture percent identity, defined by the color bar below. (c) Plaquing of the phages indicated on *E. coli* K12 harboring *cmdTAC* under control of its native promoter or an empty vector control. Data used to generate EOP data in Fig. 1c. (d) Plaquing of T4 phage on EV, *cmdTAC*[+], or *cmdT\*AC*[+]. (e) Survival assay of empty vector, *cmdTAC*[+] cells or cells expressing a direct defense system (PD-T4-1) from its native promoter on the same low-copy plasmid. Values represent the number of CFUs of each strain after 18 min of infection divided by the CFUs pre-infection. Boxplots represent the median and quartile ranges of n = 4 biological replicates, with whiskers indicating max/min values. *p*-values are indicated at the top and represent a two-sided independent t-test. (f) Efficiency of center of infection assays with *cmdTAC*[+] cells which measures the number of infected cells that go on to produce > 0 progeny phage. Value is measured relative to the empty vector control. Boxplots represent the median and quartile ranges of n = 4 biological replicates, with whiskers indicating max/min values.

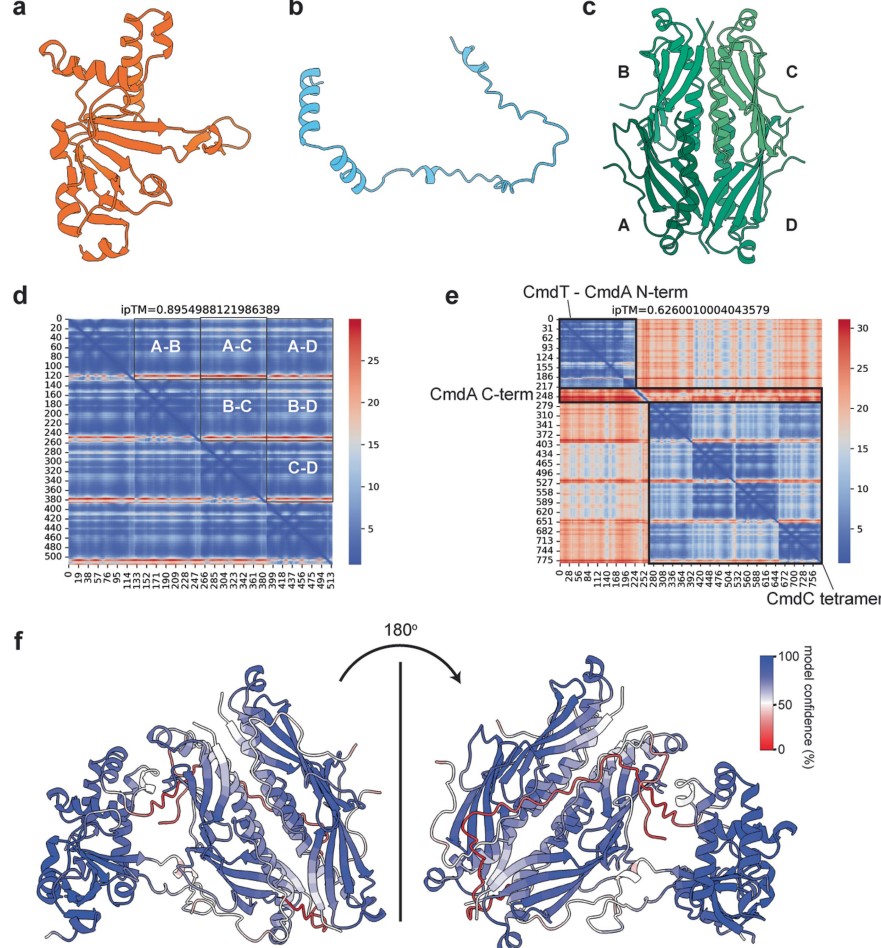

**Extended Data Fig. 2 | AlphaFold2-based prediction of CmdTAC structures.**
(a-c) AlphaFold2 predicted structures of CmdT (A), CmdA (B), and tetrameric CmdC (C). For the CmdC tetramer in panel (c) each subunit is labeled with a letter (A-D) where noted in panel (d). (d) PAE plot and per-residue model confidence score (pLDDT) for AlphaFold2-predicted CmdC tetramer. For the upper triangle of the plot, each subunit-subunit interaction is boxed and labeled with the corresponding subunit interactions corresponding to the subunits labeled in (c). (e) PAE plot and per-residue model confidence score (pLDDT) for the AlphaFold2-predicted complex of CmdTAC. Outlined in black and labeled within the plot are regions corresponding to notable features in the CmdTAC complex. (f) Ribbon diagrams of predicted CmdTAC complex, color-coded based on per-residue model confidence score (pLDDT).

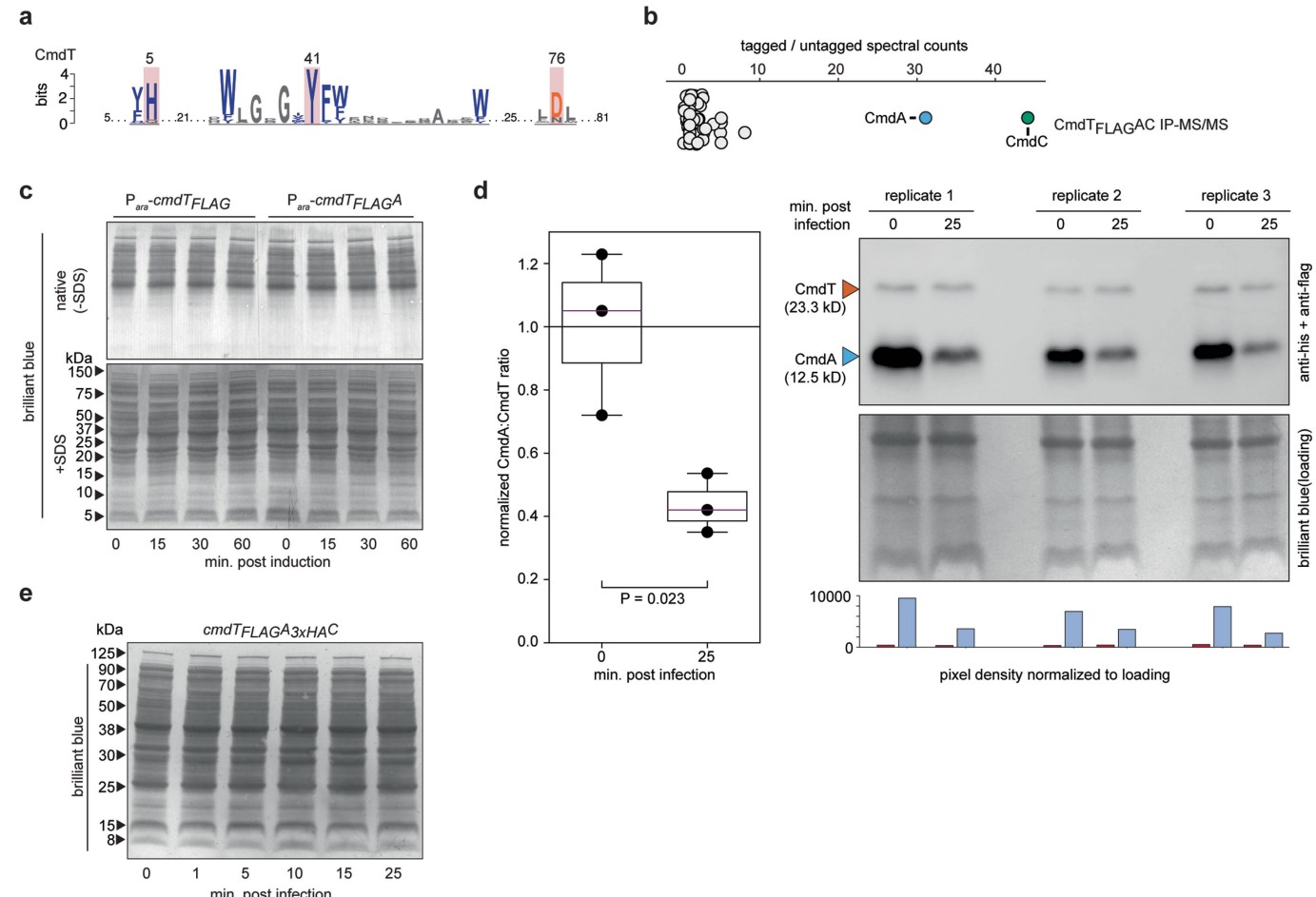

**a**

CmdT

**b**

**c**

**d**

**e**

**Extended Data Fig. 3 | Sequencing logo and IP-MS/MS interacting partners of CmdT, loading controls of for immunoblots, and quantification of CmdA degradation.** (a) Sequence logo showing conservation of putative catalytic residues in CmdT homologs. Shaded residues indicate active site residues in known ADP-ribosyltransferases. Blue color indicates conserved, aromatic residues. Amino acid numbers for CmdT[ECOR22] listed across the top. (b) IP-MS/MS of proteins co-precipitating with N-terminally FLAG-tagged CmdT. Data points corresponding to CmdC and CmdA are labeled. The x-axis indicates the ratio of spectral counts for $cmdT_{FLAG}AC$ and $cmdTAC$ containing cells, both on low copy plasmids under the $cmdTAC$ native promoter. A pseudocount is added to each sample set. (c) Coomassie stained gels used as loading controls for immunoblots shown in Fig. 2b. (d) Western blot for CmdT[FLAG] and CmdA[3xHA] at 0 and 25 min post T4 infection of ML4220. Left, quantification of CmdA to CmdT ratios as measured by pixel intensity of the Western blot on the right and normalized to total protein stain, bottom right. The normalized ratios were adjusted so that the mean time 0 value is equal to 1.0. Box plots display the values, median, and quartiles of three biological replicates, with whiskers indicating max/min values. Two-sided independent t-test $p$-value is shown. (e) Coomassie stained gel used as loading control for immunoblot shown in Fig. 2e.

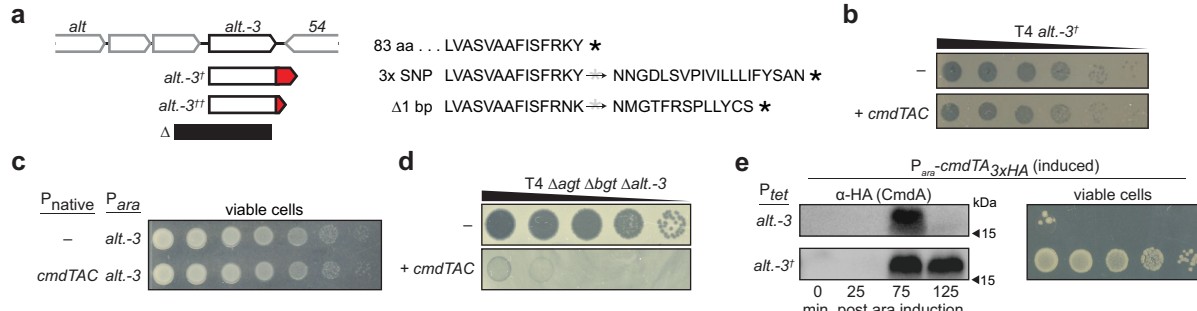

**Extended Data Fig. 4 | Mutations to *alt.-3* inhibit CmdTAC defense by blocking CmdA degradation.** (a) The *alt.-3*† and *alt.-3*†† mutations isolated as *cmdTAC* escape mutants are summarized below a diagram of the *alt.-3* region of the T4 genome. The region deleted in T4 Δ*agt* Δ*bgt* Δ*alt.-3* is shown below. The C-termini of Alt.-3, Alt.-3†, and Alt.-3†† are shown on the right. (b) Plaquing of 10-fold serially diluted T4 phage harboring the *alt.-3*† mutation on lawns of *E. coli* expressing *cmdTAC* or carrying an empty vector. (c) Plating viability of strains harboring *cmdTAC* or an empty vector and expressing the wild-type *alt.-3* gene. (d) Plaquing of 10-fold serially diluted T4 Δ*agt* Δ*bgt* Δ*alt.-3* phage on lawns of *E. coli* harboring *cmdTAC* or an empty vector. The bacterial genotype was Δ*mcrA* Δ*mcrBC*, as required for T4 Δ*agt* Δ*bgt*. (e) Cells expressing *cmdTA* from an inducible promoter with CmdA engineered to have an N-terminal 3xHA tag. Cells also express *alt.-3* or *alt.-3*† from a tetracycline-inducible promoter. CmdA levels were measured by immunblot (left), with cell viability assessed by 10-fold serial dilutions (right).

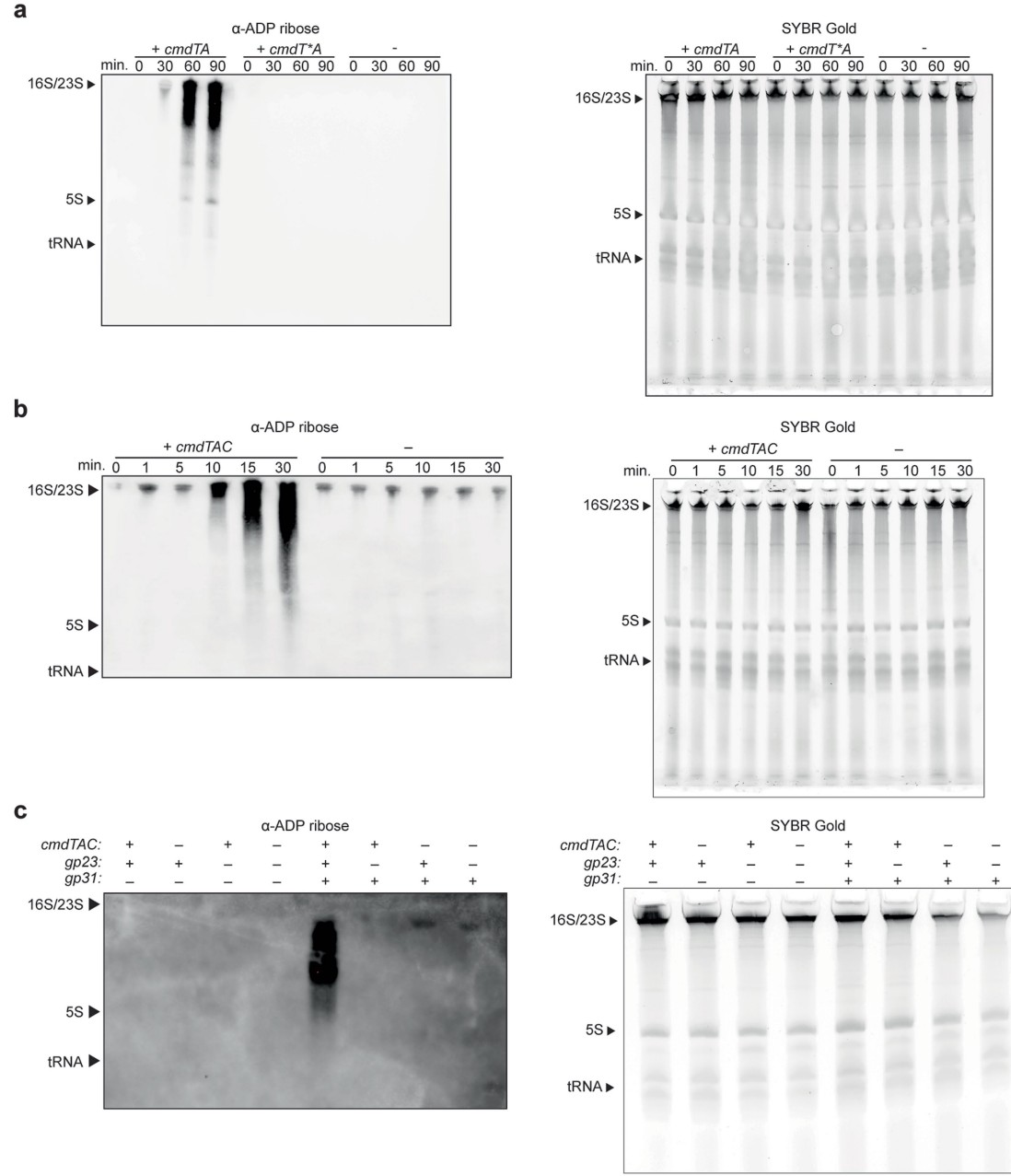

**Extended Data Fig. 5 | Anti-ADP ribose immuno-northern blots and total RNA staining of TBE-urea gels for varying methods of CmdT expression.** (a) Immuno-northern blot (left) and total RNA (right) of RNA samples taken from cells harboring arabinose-inducible *cmdTA*, *cmdT*A*, or an empty vector harvested at the times indicated post-induction. RNA was resolved on polyacrylamide gels. (b) Same as panel (a) but for cells harboring *cmdTAC* infected with T4 and harvested post-infection at the times indicated. (c) Same as panel (a) but for cells harboring *cmdTAC*, or an empty vector, and producing the combinations of Gp23 and Gp31 indicated.

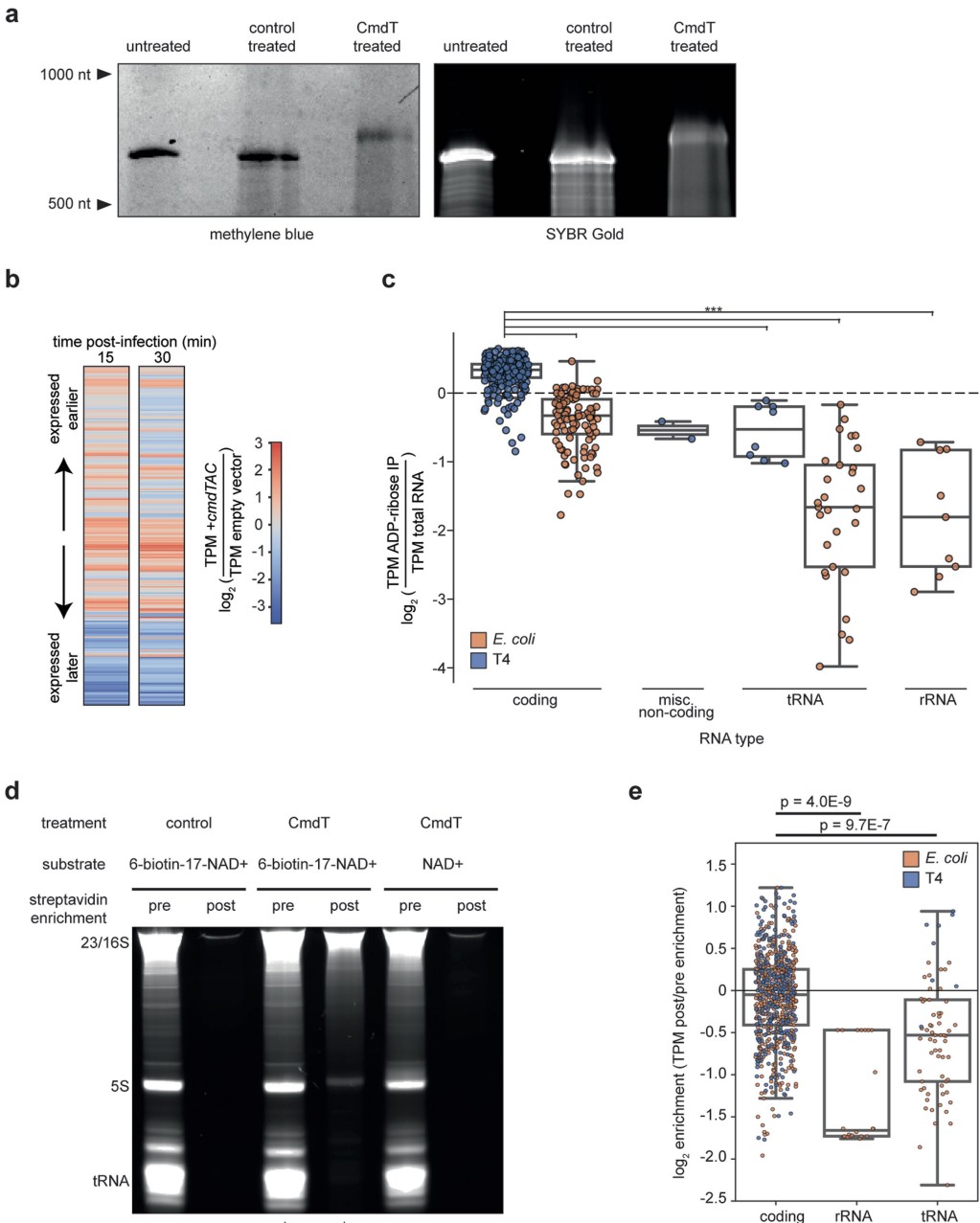

**Extended Data Fig. 6 | TBE-urea gel resolution of in vitro modified mRNA and RNA-seq and RIP-seq analysis of ADP-ribosylated RNA.** (a) TBE-urea gel showing in vitro transcribed DHFR mRNA used as a translation template for the in vitro translation conducted in Fig. 3f. The DHFR transcript was either treated with a control eluant or CmdT. CmdT-treatment resulted in molecular weight increase equivalent to approximately 80 nucleotides. Sizes were determined based on a low-range ssRNA marker. (b) Heat maps summarizing, for each T4 transcript, the ratio of TPM values in cells harboring *cmdTAC* or an empty vector. Transcripts are ordered from top to bottom based on peak time of expression, as determined previously[44]. Data are the average of two biological replicates. (c) Ratio of ADP-ribose IP enriched TPM to baseline RNA TPM values for each transcript. Transcripts are separated by RNA type and T4 vs MG1655 origin. Note that the T4 genome does not contain rRNA. Boxplots represent the median and quartile ranges, with whiskers indicating 1.5x interquartile range. Asterisks indicate p-value < 0.01 for a two-sided Welch's t-test. Exact *p*-values for tests shown (all in comparison to T4 coding transcripts) are: *E. coli* coding, *p* = 9.9E-35; T4 tRNA, *p* = 4.9E-4; *E. coli* tRNA, *p* = 1.3E-11; *E. coli* rRNA, *p* = 9.0E-5. Data are the average of n = 2 biological replicates. (d) TBE-urea gel showing total RNA from T4-infected cells after treatment +/- CmdT and with 6-biotin-17-NAD$^+$ or NAD$^+$. RNA is shown pre- and post- streptavidin enrichment, with arrows indicating samples that were sequenced. (e) Ratio of TPM after streptavidin enrichment of biotinylated RNA to TPM pre-enrichment for each transcript. Transcripts are separated by RNA type and T4 vs MG1655 origin. Note that the T4 genome does not contain rRNA. Two-sided Welch's t-test p-values are indicated. Boxplots represent the median and quartile ranges, with whiskers indicating 1.5x interquartile range.

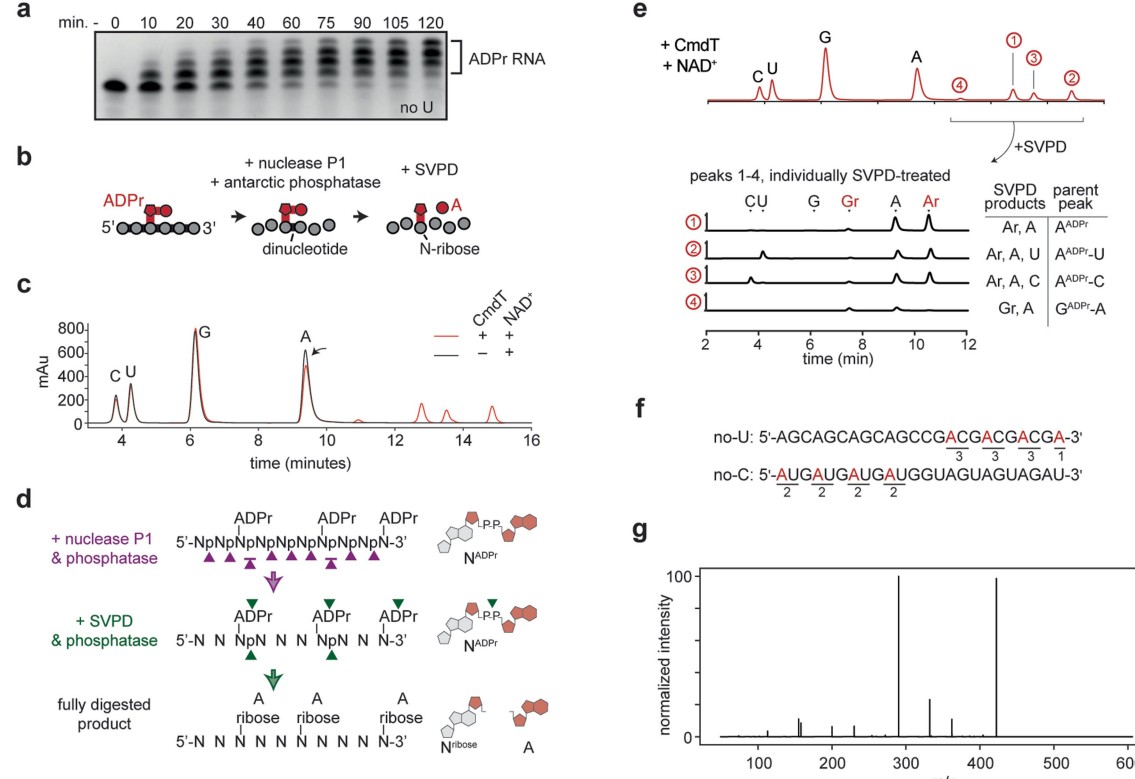

**Extended Data Fig. 7 | Additional analyses of ADP-ribosylation of RNA by CmdT.** (a) The no-U oligo (see Fig. 4d) incubated with CmdT for the times indicated, with all reactions visualized as in Fig. 4. (b) Schematic showing enzymatic digestion of ADP-ribosylated oligos with nuclease PI and antarctic phosphatase, with or without snake venom phosphodiesterase (SVPD). (c) Overlay of HPLC traces from Fig. 4f derived from analysis of nucleosides isolated following incubation of the no-U and no-C RNA oligos with CmdT and NAD$^+$ (or without NAD$^+$, as indicated) and then treated with nuclease P1 and antarctic phosphatase. Arrow highlights the decrease in adenosine for reaction containing CmdT and NAD$^+$. Peaks corresponding to A, C, G, and U nucleosides are marked. (d) Schematic summarizing enzymatic digestion activities of nuclease P1, antarctic phosphatase, and snake venom phosphodiesterase on ADP-ribosylated RNA oligos. (e-f) HPLC analysis of nucleosides produced after treating the numbered peaks from top row of Fig. 4f (reproduced at the top) with SVPD. The products are labeled and the inferred parent molecule indicated on the far right with the origin of the peaks on the oligo substrates shown in (f). Gr = G-ribose; Ar = A-ribose; ADPr = ADP-ribose. (g) MS/MS fragmentation of the A-ribose sodium adduct. Collision energy = 25 eV, as used previously[12]. Note the absence of a ribose-ribose peak at m/z = 287.

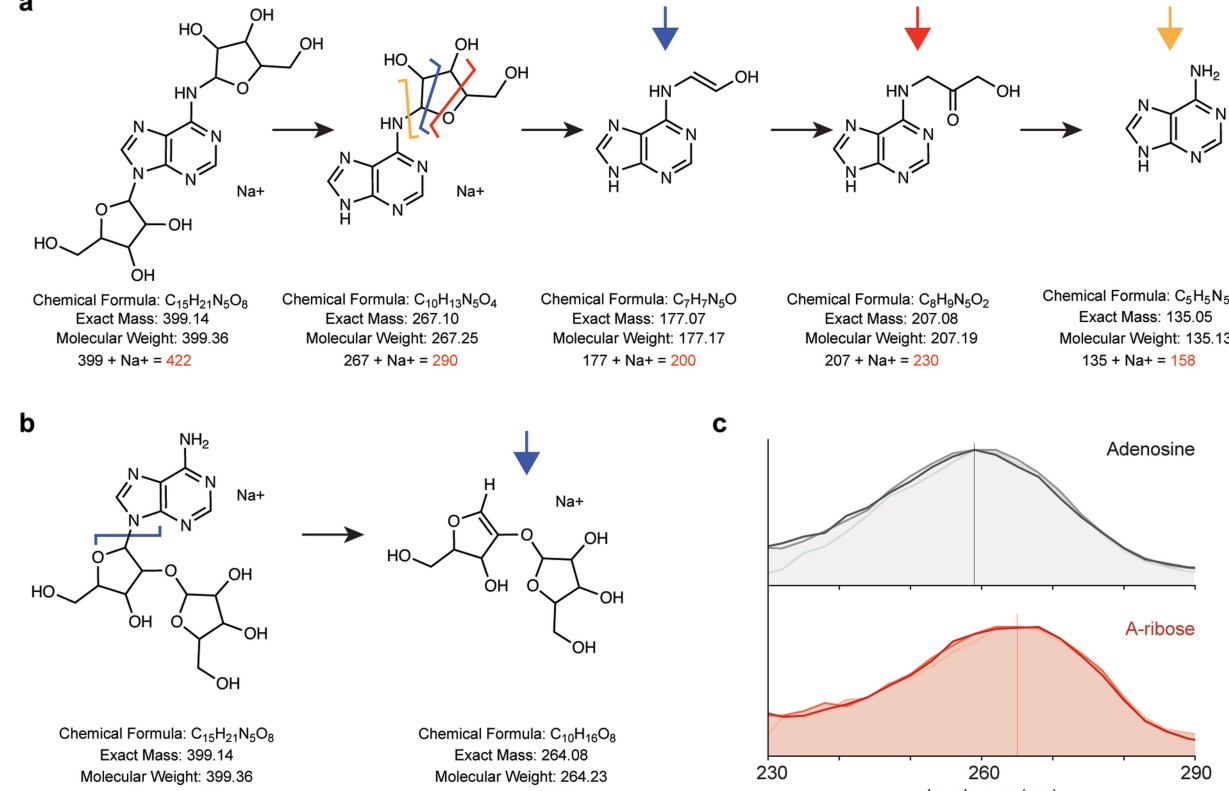

**a**

Chemical Formula: $C_{15}H_{21}N_5O_8$
Exact Mass: 399.14
Molecular Weight: 399.36
399 + Na+ = 422

Chemical Formula: $C_{10}H_{13}N_5O_4$
Exact Mass: 267.10
Molecular Weight: 267.25
267 + Na+ = 290

Chemical Formula: $C_7H_7N_5O$
Exact Mass: 177.07
Molecular Weight: 177.17
177 + Na+ = 200

Chemical Formula: $C_8H_9N_5O_2$
Exact Mass: 207.08
Molecular Weight: 207.19
207 + Na+ = 230

Chemical Formula: $C_5H_5N_5$
Exact Mass: 135.05
Molecular Weight: 135.13
135 + Na+ = 158

**b**

Chemical Formula: $C_{15}H_{21}N_5O_8$
Exact Mass: 399.14
Molecular Weight: 399.36
399 + Na+ = 422

Chemical Formula: $C_{10}H_{16}O_8$
Exact Mass: 264.08
Molecular Weight: 264.23
264 + Na+ = 287

**c**

Adenosine

A-ribose

absorbance (nm)

**Extended Data Fig. 8 | Mass spectrometry and UV-Vis analysis of CmdT-dependent ADP-ribosylated adenosine.** (a) Predicted fragmentation of adenosine ribosylated on the N6 position (far left) to produce ribosylated adenine (second from left), which is predicted to further fragment into three species (right), with m/z values of 200, 230, and 158, corresponding to the peaks seen in Fig. 4h. (b) Predicted fragmentation of adenosine ribosylated on the 2′-OH (left) to produce the ribosylated adenine (right), with an m/z value of 287. This peak was not seen in Fig. 4h, but was seen with prior analysis of RhsP2 (ref. 12). (c) UV-Vis spectroscopy of adenosine and CmdT-produced adenosine-ribose from Fig. 4f.

# Reporting Summary

## Statistics

For all statistical analyses, confirm that the following items are present in the figure legend, table legend, main text, or Methods section.

| n/a | Confirmed | |
|---|---|---|
| ☐ | ☒ | The exact sample size (*n*) for each experimental group/condition, given as a discrete number and unit of measurement |
| ☐ | ☒ | A statement on whether measurements were taken from distinct samples or whether the same sample was measured repeatedly |
| ☐ | ☒ | The statistical test(s) used AND whether they are one- or two-sided<br>*Only common tests should be described solely by name; describe more complex techniques in the Methods section.* |
| ☒ | ☐ | A description of all covariates tested |
| ☒ | ☐ | A description of any assumptions or corrections, such as tests of normality and adjustment for multiple comparisons |
| ☐ | ☒ | A full description of the statistical parameters including central tendency (e.g. means) or other basic estimates (e.g. regression coefficient) AND variation (e.g. standard deviation) or associated estimates of uncertainty (e.g. confidence intervals) |
| ☐ | ☒ | For null hypothesis testing, the test statistic (e.g. *F*, *t*, *r*) with confidence intervals, effect sizes, degrees of freedom and *P* value noted<br>*Give P values as exact values whenever suitable.* |
| ☒ | ☐ | For Bayesian analysis, information on the choice of priors and Markov chain Monte Carlo settings |
| ☒ | ☐ | For hierarchical and complex designs, identification of the appropriate level for tests and full reporting of outcomes |
| ☒ | ☐ | Estimates of effect sizes (e.g. Cohen's *d*, Pearson's *r*), indicating how they were calculated |

*Our web collection on statistics for biologists contains articles on many of the points above.*

## Software and code

Policy information about availability of computer code

| | |
|---|---|
| Data collection | Biotek Gen5 v. 3.02 for growth curve data.<br>DefenseFinder v1.3.0 for CmdT logo creation.<br>HMMER3 v3.3.2 for CmdTAC taxonomic distribution. |
| Data analysis | ChimeraX v1.7, bowtie2 v2.3.4.1, cutadapt v1.15, samtools v1.7 genomearray3, pysam v0.16.0.1, pandas v2.0.3, numpy v1.24.4, matplotlib v3.2.2, seaborn v0.10.1, scipy v1.5.0, biopython v1.77, Geneious version 2020.2.4 |

For manuscripts utilizing custom algorithms or software that are central to the research but not yet described in published literature, software must be made available to editors and reviewers. We strongly encourage code deposition in a community repository (e.g. GitHub). See the Nature Portfolio guidelines for submitting code & software for further information.

## Data

Policy information about availability of data

All manuscripts must include a data availability statement. This statement should provide the following information, where applicable:
- Accession codes, unique identifiers, or web links for publicly available datasets
- A description of any restrictions on data availability
- For clinical datasets or third party data, please ensure that the statement adheres to our policy

Summary spectra hits and raw data for IP-MS/MS of CmdC and CmdT pulldowns were deposited at MassIVE and can be accessed at doi:10.25345/C52J68G0H. Raw

data for nucleotide MS and ESI-MS/MS were deposited at MassIVE and can be accessed at doi:10.25345/C51N7XZ2Q. RNA-seq and RIP-seq data are available at GEO under accession number GSE253514.

# Research involving human participants, their data, or biological material

Policy information about studies with [human participants or human data](). See also policy information about [sex, gender (identity/presentation), and sexual orientation]() and [race, ethnicity and racism]().

| | |
|---|---|
| Reporting on sex and gender | N/A |
| Reporting on race, ethnicity, or other socially relevant groupings | N/A |
| Population characteristics | N/A |
| Recruitment | N/A |
| Ethics oversight | N/A |

Note that full information on the approval of the study protocol must also be provided in the manuscript.

# Field-specific reporting

Please select the one below that is the best fit for your research. If you are not sure, read the appropriate sections before making your selection.

☒ Life sciences  ☐ Behavioural & social sciences  ☐ Ecological, evolutionary & environmental sciences

For a reference copy of the document with all sections, see [nature.com/documents/nr-reporting-summary-flat.pdf]()

# Life sciences study design

All studies must disclose on these points even when the disclosure is negative.

| | |
|---|---|
| Sample size | All experiments were performed in at least triplicate with the exception of RNA-seq, RIP-seq, and radiolabel incorporation experiments which were performed in duplicate and IP-MS/MS experiment which were performed once. The majority of experiments were performed in triplicate because of the large effect sizes as a means of identifying reproducibility. RNA sequencing based experiments were performed in duplicate both due to the high cost of these experiments and the large amounts of data produced through these methods. Radiolabel incorporation experiments were performed in duplicate due to the large effect size and inherent qualitative nature of the result. IP-MS/MS experiments were not repeated due to their high time and monetary costs but results were verified through independent means (i.e. activation of CmdTAC by Gp23 as shown in Fig. 2i and Extended Data Fig. 5c. |
| Data exclusions | No data were excluded. |
| Replication | All experimental findings were repeated at least twice. All report results were successfully reported. |
| Randomization | All experiments were performed in isogenic strains so there were no covariates to control for. No subjective choice of experimental and control groups was performed. |
| Blinding | Blinding was not relevant because all data were discrete and/or raw data is reported in the manuscript |

# Reporting for specific materials, systems and methods

We require information from authors about some types of materials, experimental systems and methods used in many studies. Here, indicate whether each material, system or method listed is relevant to your study. If you are not sure if a list item applies to your research, read the appropriate section before selecting a response.

## Materials & experimental systems

| n/a | Involved in the study |
|-----|-----------------------|
| ☐ | ☒ Antibodies |
| ☒ | ☐ Eukaryotic cell lines |
| ☒ | ☐ Palaeontology and archaeology |
| ☒ | ☐ Animals and other organisms |
| ☒ | ☐ Clinical data |
| ☒ | ☐ Dual use research of concern |
| ☒ | ☐ Plants |

## Methods

| n/a | Involved in the study |
|-----|-----------------------|
| ☒ | ☐ ChIP-seq |
| ☒ | ☐ Flow cytometry |
| ☒ | ☐ MRI-based neuroimaging |

## Antibodies

| Antibodies used | Poly/Mono-ADP Ribose (E6F6A) Rabbit mAb, Cell Signaling Technologies, Cat #: 83732<br>6x-His Tag Monoclonal Antibody (HIS.H8), Thermo Fisher, Cat #: MA1-21315<br>HA-Tag (C29F4) Rabbit mAb, Cell Signaling Technologies, Cat #: 3724S,<br>DYKDDDDK Tag (D6W5B) Rabbit mAb, Cell Signaling Technologies, Cat #: 14793S<br>Goat anti-Mouse IgG (H+L) Secondary Antibody, HRP, Thermo Fisher, Cat #: 32430<br>Goat anti-Rabbit IgG (H+L) Secondary Antibody, HRP, Thermo Fisher, Cat #: 32460 |
|---|---|
| Validation | Antibodies have been validated by manufacturer as listed on website or in listed publications:<br>Poly/Mono-ADP Ribose: Bullen et al., An ADP-ribosyltransferase toxin kills bacterial cells by modifying structured non-coding RNAs Mol. Cell (2022) DOI:https://doi.org/10.1016/j.molcel.2022.08.015<br>anti-His: from manufacturer "verified by Relative expression to ensure that the antibody binds to the antigen stated"<br>anti-HA: validated by "Western blot analysis of extracts from HeLa cells, untransfected or transfected with either HA-FoxO4 or HA-Akt3, using HA-Tag (C29F4) Rabbit mAb.<br>anti-DYKDDDDK: validated by "Western blot analysis of extracts from 293T cells, mock transfected or transfected with DYKDDDDK-GFP.<br>anti-Mouse: validated by "Western blot analysis of HA Epitope Tag performed by various amounts of E. coli lysate containing a multi-epitope tagged protein.<br>anti-Rabbit: validated by "Western blot analysis performed on membrane enriched extracts of K562 and PC-3." |

## Plants

| Seed stocks | *Report on the source of all seed stocks or other plant material used. If applicable, state the seed stock centre and catalogue number. If plant specimens were collected from the field, describe the collection location, date and sampling procedures.* |
|---|---|
| Novel plant genotypes | *Describe the methods by which all novel plant genotypes were produced. This includes those generated by transgenic approaches, gene editing, chemical/radiation-based mutagenesis and hybridization. For transgenic lines, describe the transformation method, the number of independent lines analyzed and the generation upon which experiments were performed. For gene-edited lines, describe the editor used, the endogenous sequence targeted for editing, the targeting guide RNA sequence (if applicable) and how the editor was applied.* |
| Authentication | *Describe any authentication procedures for each seed stock used or novel genotype generated. Describe any experiments used to assess the effect of a mutation and, where applicable, how potential secondary effects (e.g. second site T-DNA insertions, mosiacism, off-target gene editing) were examined.* |

