## [Peer Review File · Nature]

Manuscript Title: Anti-viral defense by an mRNA ADP-ribosyltransferase that blocks translation

Reviewer Comments & Author Rebuttals

Reviewer Reports on the Initial Version:

Referees' comments:

Referee #1 (Remarks to the Author):

The manuscript by Vassallo et al characterises the antiphage activity of a chaperone-dependent toxin-antitoxin system, CmdTAC from *E. coli*. There are an increasing number of toxin-antitoxin systems known to have impact in phage defence, but this is the first example for a chaperone-dependent system having this activity. Through a set of elegant biochemical analyses the authors deduce the ADP-ribosyltransferase activity of the CmdT toxin. It appears to target GA pairs in free mRNA, the first time an ADP-ribosyltransferase has been noted to target mRNA to the knowledge of the authors (and this reviewer).

The data add to the field of phage defence, toxin-antitoxin systems, and potentially opens up exploration of similar activities in eukaryotic immunity. It should be of interest to a wide audience. Overall the manuscript is robust, but there are some parts that need further evidence.

As an aside – please put line numbers in the manuscript to help your reviewers.

Specific comments:

Last para intro – typo for ref21

First line results – please explain how cmdTAC was discovered

First para results – Efficiency of Plating not plaquing

First para results and throughout – no detail is given to the plasmids used. Please give specifics, not simply “a plasmid containing cmdTAC”. I would ask this change throughout the manuscript for consistency of approach and allows the reader to know exactly what is being used.

First para results – “CmdTAC strongly decreased” please give more specific detail

First para page 5 – my major gripe with the paper – last line is woefully insufficient evidence to claim Abi activity. You need more – evidence of cell death, Efficiency of Centre of Infection experiments.

Second para page 5 – “predicted a structure with similarity” is nebulous. How is this judged, give specifics. Please give details of the predicted complex stoichiometries. The complex actually looks pretty poor with few interactions between CmdC and TA – perhaps run PISA analyses of interactions. Do you have any experimental evidence that these three interact or is it all inferred? We need to be wary of simply publishing AFold outputs without validation.

Third para page 7 – give plasmid details for inducible plasmid. Third para page 7 – you do not transform a plasmid, you transform a strain with a plasmid. Please correct. A delatclpP strain of what? Please be specific. This goes back to lack of detail on strain and plasmid usage. Often throughout the manuscript a variant of “a strain harboring xxx” is used. How about simply stating what is used for clarity and precision?

Last line of page 7 – what is original size?

Third para page 8 – what is mechanism of A stabilisation? Binding? ClpP inhibition? (I see this appears later in the disc so maybe can ignore this comment)

Last line page 8 – you imply what strain but do not specify where the samples actually come from. Please do so.

Second para page 9 – first line is not supported yet no evidence of Abi.

Third para – how were the electrostatics generated?

Figure legends generally lack detail; they do not specify strains used, numbers of replicates, or even cursory experimental detail. As a result, they leave a lot to assumption. Please expand substantially.

Fig. 1F – state that inset of CmdC is for a single protomer. How were RMSD values generated? The given RMSD values are actually rather poor.

Fig. 1E – log scale please

Fig. 2 – re-arrange panels to go in order

Fig. 3D – explain genotypes of T4 derivative and corresponding E coli host

Fig. 3G – log scale please. Can we see the Gp23 only result here too please?

Fig. 5A – odd labelling at the top – is the hyphen an empty control lane? If so, it looks cropped off

Referee #1 (Remarks on code availability):

Not qualified to review the code

Referee #2 (Remarks to the Author):

Toxin-antitoxin (TA) systems are found in many bacterial genomes where they play an important role in defense against invading bacteriophage. The antitoxin usually forms a complex with the toxin and thereby neutralizes it. Degradation of the antitoxin then allows the toxin to become active to block cell growth and/or mediate cell death. In TAC systems, an additional chaperone is necessary to stabilize the antitoxin. The past years have seen tremendous developments in our understanding of the diverse ways that TA and TAC systems operate. Here the authors discover a TAC system, CmdTAC, where the toxin, CmdT, is an ADP-ribosyltransferase that specifically modifies mRNAs to block phage translation and production of mature virions. This is exciting since ADP-ribosyltransferases are better known for modification of proteins. However, it must be noted that they have also been shown, as in the case of the DarTG TA-system, to provide phage defense by ADP-ribosylating the guanines of viral DNA. Nevertheless, in this study, Vassallo and coworkers find that CmdT acts by ADP-ribosylate the N6 of adenines in “GA” dinucleotides within mRNAs i.e. not protein or DNA.

Another important advance arising from the manuscript is that the authors provide insight into how the toxin is activated in response to phage infection, namely, that the major capsid protein outcompetes the antitoxin CmdA for binding to the chaperone CmdC, leading to degradation of CmdA by ClpP and thereby activation of CmdT.

Overall, I find this manuscript presents a very complete story from discovery to mechanism. As far as I can judge, the experiments are well-performed and appropriately interpreted. I don't feel that that the manuscript requires additional experiments but I have nevertheless outlined a few points that were not fully clarified.

Minor points.

1. It was not clear to me why CmdTAC does not affect Bas35 and Bac38? Does this pertain to some mechanism to overcome CmdTAC...I guess the authors already looked at the obvious ones that they identified in their study?
2. The C-terminal extension to Alt-3 inhibits CmdA degradation. It was unclear to me whether it is simply the 110-117 amino acids that are necessary for this or whether it is the protein with the extension? This would be easy to clarify.
3. Ultimately, the mechanism of CmdT is to inhibit translation of the phage (and host) mRNAs. The authors discuss about possibility of modification of Shine-Dalgarno regions of the mRNA as well as GA present in the reading frame. Given the size of the modification, one can easily imagine that ADP-ribosylation would indeed prevent the mRNA binding to the ribosome if the GA in the SD sequence is modified, or prevent mRNA decoding and translocation if the GA is in the reading frame. Maybe it's a small point, but if the SD is modified, would not one expect a dramatic decrease in the association of these mRNAs with ribosomes which could be seen in the Ribo-seq data? Also since many (actually most) E.coli proteins don't even have a SD sequence, presumably the authors already have such candidates where association is not lost but translation inhibition is still observed, which could be given as examples. Ultimately, the N6 of adenine which is modified would prevent any sort of Watson-crick base-pairing...which should at least be mentioned in the discussion (unless I missed it).
4. I was a little surprised that the authors didn't cite the recent work in Nature about a viral ADP-ribosyltransferase that attaches RNA chains to host proteins, e.g. Wolfram-Schauerte, et al Nature 620 (7976), pp. 1054 - 1062 (2023).

Referee #3 (Remarks to the Author):

This is an exciting study that describes characterises the CmdTAC system and its mechanism in protecting bacteria from infection by phages from the T-even family. Whilst previous mechanistic work has already shed light onto the CmdTAC system, several important questions had remained unresolved. In particular, the authors demonstrate that the CmdC subunit of CmdTAC senses the viral capsid protein Gp23, which gives rise to de-repression of the toxic CmdT subunit of the complex. The authors further reveal that CmdT displays mRNA-directed ADP-ribosyltransferase activity to preferentially block the translation of viral transcripts. The work encompasses an impressive array of techniques, including bacterial genetics, structure prediction, bioinformatics, proteomics, transcriptomics and mass spectrometry. The authors elegantly determine the ADP-ribose modification site as N6 of adenine within GA dinucleotides.

CmdT appears to be the first example of an mRNA-directed ART. The study is of broad relevance to the field of antiviral immunity and opens up several exciting research avenues for subsequent studies. The work is of high quality and mostly well-presented.

I have a number of points which should be addressed:

- In Figure 2B, left hand panel, it is not clear to me why protein expression increases at 15 min, then decreases, only to increase again. Is this pattern reproducible or stochastic?
- Figure 2B: The use of the native gel is initially not clear (p. 6, paragraph 1). The authors may want

to introduce its purpose upfront.

- p. 6, paragraph 2: If CmdA is indeed needed for the production of a stable CmdT and then removed, how do the authors explain that non-aggregated CmdT is present after the 25-min mark post-induction (Figure 2B), when CmdA is nearly undetectable (Figure 2C)?
- Figure 2E: A quantification, ideally of several independent experiments, would be desirable.
- Figure 2F: I assume the "WT" strain lacks CmdTAC? This would be consistent with the earlier data in the manuscript. If this is the case, can the figure be labelled more clearly? This applies to multiple figures in the manuscript.
- Figure 2: Can the order of panels be made more intuitive? Would it be worth to include loading controls for the western blots in the main figures throughout the manuscript? I realise these are provided as part of the supplementary material. Where molecular weight markers are missing from figures, can these please be added?
- It would be reassuring to see a co-immunoprecipitation analysis of CmdC with Gp23 (and/or Gp21).
- p. 7, paragraph 4: "CmdTAC likely forms a stable complex in the absence of phage, with infection then triggering the ClpP-dependent degradation of CmdA, release of CmdT ..." - This hypothesis could also be tested by co-immunoprecipitation.
- Figure 3G: Is a growth curve for cmdTAC only available, for side-by-side comparison? I can see this has been done for Figure 1E, but a parallel control experiment for Figure 3G would have been helpful.
- Figure 4A: Speculating about the identity of these bands may be secondary as the signal observed may be non-specific. Can a positive control (e.g., mono- and poly-ADP-ribose) be included? As a minor query, why did the authors choose this antibody rather than other ADP-ribose detection reagents (e.g., https://www.merckmillipore.com/GB/en/product/Anti-pan-ADP-ribose-binding-reagent,MM_NF-MABE1016)?
- Figures 5B and C need molecular weight markers. It would be helpful to document mRNA ADP-ribosylation for Figure 5C in the supplementary material.
- Figure S2: In the PAE plots, can the authors highlight the domains and modules shown in panels A, B and C, for example by boxes? For the text (p. 5, paragraph 2), it may be worth pointing out that the extended C-terminus of CmdA is only predicted with low confidence, as apparent from the pLDDT score.
- p. 3, paragraph 1: The term "poly-ADP-ribosyl polymerase" is outdated. Instead of "polymerase", we are encouraged to use the term "transferase", given that polymerases depend on templates whilst transferases do not (reference: PMID 34323016).

- Details of mass spec data should be provided.
- RNA-DNA hybrids can also serve as substrates of reversible ADP-ribosylation. It would be interesting to examine the ADP-ribosylation of R-loop structures.
- What does the alt-3 gene code for? Can a brief description of its function be added? Do the identified mutant variants represent loss or gain of function? What phenotype do the authors expect when alt-3 is deleted? Non-experts in phage biology will have difficulties to follow.
- What portion of translation inhibition is due to host endoribonucleases and viral mRNA degradation?
- Can the authors speculate why a phage defense system would be toxic to the bacterial cells themselves? Later data show that host mRNAs can also be ADP-ribosylated. Also, the double-edged sword of the defense system protecting the cells on the one hand and killing them on the other (in the experimental setting) could be introduced more explicitly, early on in the manuscript. Readers without an expert background in this field might get confused by the two sets of assays (viral plaque formation and viability).
- Do the authors have direct biochemical evidence that cmdT* is catalytically inactive? Numerous ARTs featuring an HYE/D triad display poly-ADP-ribosyltransferase activity. I assume the CmdT is a mono-ART, but have the authors explored the nature of the modification more extensively?
- Can the authors explore whether ADP-ribose hydrolysis reverses the effect of cmdT?
- Can the authors provide suggestions for how CmdA might block CmdT function?

General comment: Can figure legends please be provided right underneath the figures? This would help the reviewers with assessing the work. Thank you!

Referee #3 (Remarks on code availability):

n.a.

Author Rebuttals to Initial Comments:

We thank the reviewers for their enthusiasm for our work and for their constructive comments and suggestions for improving the manuscript. Below we respond to each query, indicating how the text and figures have been modified. A 'Track Changes' version of the manuscript as well as a 'clean' version are both provided in the submitted materials, but please note that line numbers indicated below refer to those in the 'clean' version - for reasons we cannot explain or control, Microsoft Word introduces unusual gaps in the line numbering of a marked up file.

Referee #1 (Remarks to the Author):

The manuscript by Vassallo et al characterises the antiphage activity of a chaperone-dependent toxin-antitoxin system, CmdTAC from *E. coli*. There are an increasing number of toxin-antitoxin systems known to have impact in phage defence, but this is the first example for a chaperone-dependent system having this activity. Through a set of elegant biochemical analyses the authors deduce the ADP-ribosyltransferase activity of the CmdT toxin. It appears to target GA pairs in free mRNA, the first time an ADP-ribosyltransferase has been noted to target mRNA to the knowledge of the authors (and this reviewer).

The data add to the field of phage defence, toxin-antitoxin systems, and potentially opens up exploration of similar activities in eukaryotic immunity. It should be of interest to a wide audience. Overall the manuscript is robust, but there are some parts that need further evidence.

As an aside – please put line numbers in the manuscript to help your reviewers.

Line numbers have been added.

Specific comments:

1. Last para intro – typo for ref21

Corrected.

2. First line results – please explain how cmdTAC was discovered

We have added a short description of the discovery of *cmdTAC* (lines 72-73).

3. First para results – Efficiency of Plating not plaquing

Corrected (lines 77-78).

4. First para results and throughout – no detail is given to the plasmids used. Please give specifics, not simply “a plasmid containing cmdTAC”. I would ask this change throughout the manuscript for consistency of approach and allows the reader to know exactly what is being used.

We thank the reviewer for highlighting these issues of clarity, which we have now tried to address. We prefer not to use strain and plasmid names that are arbitrary and unfamiliar to the reader, and instead have provided the key details in the text (with corresponding strain/plasmid names and genotypes provided in the Methods and strain/plasmid tables). For example, rather than stating that a strain is *E. coli* MG1655 + pCD2, we write that we cloned *cmdTAC* under control of its native promoter on a low-copy plasmid in the wild-type strain. We think the latter is most accessible to non-experts. Strain and plasmid details have also been added to the figure legends to facilitate interpretation of the data presented. Finally, we have tried, per the reviewer's request, to be consistent in our strain and plasmid descriptions throughout to avoid confusion. If there are any specific passages of the text that need further modification or clarification, we are happy to amend them.

5. First para results – “CmdTAC strongly decreased” please give more specific detail

We have now included a more specific description of CmdTAC defense in lines 82-83.

6. First para page 5 – my major gripe with the paper – last line is woefully insufficient evidence to claim Abi activity. You need more – evidence of cell death, Efficiency of Centre of Infection experiments.

We have now included both ECOI (0.036) and survival (not significantly different from empty vector) assays (see new Fig. S1e-f and lines 92-99), which are consistent with an abortive infection mechanism.

7. Second para page 5 – “predicted a structure with similarity” is nebulous. How is this judged, give specifics. Please give details of the predicted complex stoichiometries.

We used Foldseek of the AlphaFold models to predict proteins with similar structures and all comparisons are made to the top Foldseek hit for each structure. This is now stated in the main text (lines 103-105) and Methods (lines 1119-1121). With regard to stoichiometries, we now explicitly note that the AlphaFold prediction is for a 1:1:4 complex of CmdT:CmdA:CmdC, in line with prior work on other TAC systems showing that their chaperones adopt a tetrameric structure like SecB.

The complex actually looks pretty poor with few interactions between CmdC and TA – perhaps run PISA analyses of interactions. Do you have any experimental evidence that these three interact or is it all inferred? We need to be wary of simply publishing AFold outputs without validation.

As we show later in the Results section (lines 218-220 and Figure 3f), an IP of CmdC-FLAG followed by mass spectrometry identified CmdT and CmdA, provides further evidence that the three components form a complex. We also separately conducted IP-MS/MS on CmdT, which pulls down CmdA and CmdC. We have now added the latter result, *i.e.* the CmdT IP-MS

experiment, as Figure 1h (and lines 120-123) to provide experimental evidence for the interactions alongside the AlphaFold prediction.

As the reviewer notes, although there is strong predicted interaction between CmdA and CmdT and within the CmdC tetramer, the interaction between CmdTA and CmdC largely hinges on the low-confidence predicted C-terminus of CmdA. While this is not entirely surprising given that antitoxins (especially of TAC systems) commonly have large unstructured regions, we have updated the text to better reflect the confidence of the AlphaFold prediction. However, we think the addition of the CmdT IP-MS experiment, along with the CmdC IP-MS experiment that was already in the paper, bolsters our argument that CmdTAC forms a complex.

8. Third para page 7 – give plasmid details for inducible plasmid.

Corrected.

Third para page 7 – you do not transform a plasmid, you transform a strain with a plasmid. Please correct.

Corrected.

A delatclpP strain of what? Please be specific. This goes back to lack of detail on strain and plasmid usage.

We now state that it is $\Delta clpP$ in MG1655 (WT).

Often throughout the manuscript a variant of “a strain harboring xxx” is used. How about simply stating what is used for clarity and precision?

Please see our response to query #4 above.

9. Last line of page 7 – what is original size?

We appreciate the reviewer catching this oversight; we have now included the size of WT Alt.-3 (96 amino acids) in line 186.

10. Third para page 8 – what is mechanism of A stabilisation? Binding? ClpP inhibition? (I see this appears later in the disc so maybe can ignore this comment)

To try and address this issue, we constitutively expressed a GFP construct fused to a linker and *ssrA* tag (which promotes rapid turnover of the GFP construct) alongside tetracycline-inducible Alt.-3, Alt.-3[†], or an empty vector control. Expression of Alt.-3[†] caused a discernable increase in GFP levels as assessed by Western blotting (see data directly below) suggesting that Alt.-3[†] prevents CmdA degradation by generally inhibiting ClpP degradation. However, Alt.-3 also very slightly increased GFP levels, albeit not to the same degree as Alt.-3[†]. These observations

suggest that wild-type Alt.-3 is a poor substrate for ClpP but to some degree blocks the degradation of other ClpP substrates if overproduced. The additional C-terminal extension in Alt.-3[†] then likely increases the inhibition of ClpP enough to prevent CmdA degradation and allow for subsequent T4 escape. We opted not to include these data in the paper as they do not definitely prove how Alt.-3[†] blocks CmdA turnover and we felt that it would disrupt the flow of the paper which is, of course, primarily focused on the mechanism of CmdTAC activation and CmdT function. We do, as noted by the reviewer, briefly speculate on Alt.-3[†] in the Discussion.

11. Last line page 8 – you imply what strain but do not specify where the samples actually come from. Please do so.

We have now included the plasmid information in the figure legend.

12. Second para page 9 – first line is not supported yet no evidence of Abi.

Please see our response to comment #6 above regarding additional evidence of Abi now added to the paper.

13. Third para – how were the electrostatics generated? Add to methods.

The electrostatics were generated with the ChimeraX coulombic function, which is now stated in the methods (lines 1119-1121).

14. Figure legends generally lack detail; they do not specify strains used, numbers of replicates, or even cursory experimental detail. As a result, they leave a lot to assumption. Please expand substantially.

We have now tried to ensure that the legends include strains, replicates for all quantified data, and other necessary experimental detail.

15. Fig. 1F – state that inset of CmdC is for a single protomer.

This is now stated in the legend for Fig. 1f.

How were RMSD values generated? The given RMSD values are actually rather poor.

We now state (line 510-511 in Fig. 1f legend) that RMSD values were generated by Foldseek. We find that Foldseek generally gives higher RMSD values than are calculated in other programs such as FATCAT and DALI. However, we report the values from Foldseek as that is what we used to identify the closest structurally similar proteins. Additionally, we expected a relatively poor RMSD between CmdT and ExoA because they are quite distantly related and likely only share a similar structure around the catalytic beta strands. Again, this was the top hit from Foldseek, which is why we show it in the figure.

16. Fig. 1E – log scale please

We have now changed the plot to log scale on the y-axis.

17. Fig. 2 – re-arrange panels to go in order

The panels of figure 2 have now been rearranged to go in order.

18. Fig. 3D – explain genotypes of T4 derivative and corresponding E coli host

These are now explained in the main text (lines 195-196) and legend of Fig. 3d. See methods (lines 1020-1024) for details of why these strains were used.

19. Fig. 3G – log scale please. Can we see the Gp23 only result here too please?

We have now changed the plot to log scale on the y-axis and included the Gp23 only result in Figure 3g.

20. Fig. 5A – odd labelling at the top – is the hyphen an empty control lane? If so, it looks cropped off

We apologize for the misleading labeling and have now corrected the issue.

Referee #2 (Remarks to the Author):

Toxin-antitoxin (TA) systems are found in many bacterial genomes where they play an important role in defense against invading bacteriophage. The antitoxin usually forms a complex with the toxin and thereby neutralizes it. Degradation of the antitoxin then allows the toxin to become active to block cell growth and/or mediate cell death. In TAC systems, an additional chaperone is necessary to stabilize the antitoxin. The past years have seen tremendous developments in our understanding of the diverse ways that TA and TAC systems operate. Here the authors discover a TAC system, CmdTAC, where the toxin, CmdT, is an ADP-

ribosyltransferase that specifically modifies mRNAs to block phage translation and production of mature virions. This is exciting since ADP-ribosyltransferases are better known for modification of proteins. However, it must be noted that they have also been shown, as in the case of the DarTG TA-system, to provide phage defense by ADP-ribosylating the guanosines of viral DNA. Nevertheless, in this study, Vassallo and coworkers find that CmdT acts by ADP-ribosylate the N6 of adenines in "GA" dinucleotides within mRNAs i.e. not protein or DNA. Another important advance arising from the manuscript is that the authors provide insight into how the toxin is activated in response to phage infection, namely, that the major capsid protein outcompetes the antitoxin CmdA for binding to the chaperone CmdC, leading to degradation of CmdA by ClpP and thereby activation of CmdT.

Overall, I find this manuscript presents a very complete story from discovery to mechanism. As far as I can judge, the experiments are well-performed and appropriately interpreted. I don't feel that the manuscript requires additional experiments but I have nevertheless outlined a few points that were not fully clarified.

Minor points.

1. It was not clear to me why CmdTAC does not affect Bas35 and Bac38? Does this pertain to some mechanism to overcome CmdTAC...I guess the authors already looked at the obvious ones that they identified in their study?

Yes, in fact we have identified a counter-defense gene, present in Bas35 but none of the others that blocks the activity of CmdTAC. When deleted from Bas35, the mutant phage can no longer escape CmdTAC defense. We are actively working on determining a mechanism. For Bas38, we are unsure, but it likely involves a separate inhibitor.

2. The C-terminal extension to Alt-3 inhibits CmdA degradation. It was unclear to me whether it is simply the 110-117 amino acids that are necessary for this or whether it is the protein with the extension? This would be easy to clarify.

Please see our response to Reviewer 1, query 10 on the same question.

3. Ultimately, the mechanism of CmdT is to inhibit translation of the phage (and host) mRNAs. The authors discuss about possibility of modification of Shine-Dalgarno regions of the mRNA as well as GA present in the reading frame. Given the size of the modification, one can easily imagine that ADP-ribosylation would indeed prevent the mRNA binding to the ribosome if the GA in the SD sequence is modified, or prevent mRNA decoding and translocation if the GA is in the reading frame. Maybe it's a small point, but if the SD is modified, would not one expect a dramatic decrease in the association of these mRNAs with ribosomes which could be seen in the Ribo-seq data? Also since many (actually most) E.coli proteins don't even have a SD sequence, presumably the authors already have such candidates where association is not lost but translation inhibition is still observed, which could be given as examples. Ultimately, the N6 of adenine which is modified would prevent any sort of Watson-crick base-pairing...which should at least be mentioned in the discussion (unless I missed it).

We have not yet performed Ribo-seq (we only present RIP-seq in the manuscript), which does not report on whether the modified RNAs are or are not associated with ribosomes. We agree with the reviewer that Ribo-seq would be a great way to pursue this question and to compare the effects of CmdT on mRNAs with and without SD sequences, though we think this is beyond the scope of the current work. As for whether ADP-ribosylation of the N6 position of adenine impacts Watson-Crick base pairing, we think it may be too speculative to comment on this in the Discussion. On one hand, N6 is not involved in hydrogen bonding with thymines and there are other modifications of the N6 position, such as methylation, that do not impact Watson-Crick base pairing. However, ADP-ribose is a much larger modification than a methylation and so could impact pairing. Again, we think this is most appropriately pursued in future studies.

4. I was a little surprised that the authors didn't cite the recent work in Nature about a viral ADP-ribosyltransferase that attaches RNA chains to host proteins, e.g. Wolfram-Schauerte, et al Nature 620 (7976), pp. 1054 - 1062 (2023).

This is now cited in the Discussion.

Referee #3 (Remarks to the Author):

This is an exciting study that describes characterises the CmdTAC system and its mechanism in protecting bacteria from infection by phages from the T-even family. Whilst previous mechanistic work has already shed light onto the CmdTAC system, several important questions had remained unresolved. In particular, the authors demonstrate that the CmdC subunit of CmdTAC senses the viral capsid protein Gp23, which gives rise to de-repression of the toxic CmdT subunit of the complex. The authors further reveal that CmdT displays mRNA-directed ADP-ribosyltransferase activity to preferentially block the translation of viral transcripts. The work encompasses an impressive array of techniques, including bacterial genetics, structure prediction, bioinformatics, proteomics, transcriptomics and mass spectrometry. The authors elegantly determine the ADP-ribose modification site as N6 of adenine within GA dinucleotides. CmdT appears to be the first example of an mRNA-directed ART. The study is of broad relevance to the field of antiviral immunity and opens up several exciting research avenues for subsequent studies. The work is of high quality and mostly well-presented.

I have a number of points which should be addressed:

1. In Figure 2B, left hand panel, it is not clear to me why protein expression increases at 15 min, then decreases, only to increase again. Is this pattern reproducible or stochastic?

This pattern is indeed reproducible. Below we show another example, conducted with slightly different parameters (30 °C) but which has the same pattern. This pattern likely reflects the misfolding of CmdT in the absence of CmdA. As CmdT is initially produced without CmdA, it is misfolded and most of it subject to degradation, leading to the initial decrease observed;

however, over time, with constant expression some misfolded CmdT that cannot be degraded may slowly accumulate, leading to the low levels seen at later time points.

2. Figure 2B: The use of the native gel is initially not clear (p. 6, paragraph 1). The authors may want to introduce its purpose upfront.

We have now changed the order and description of the native gel experiment to clarify (see lines 132-139).

3. p. 6, paragraph 2: If CmdA is indeed needed for the production of a stable CmdT and then removed, how do the authors explain that non-aggregated CmdT is present after the 25-min mark post-induction (Figure 2B), when CmdA is nearly undetectable (Figure 2C)?

Although CmdA is undetectable, it is still being produced constitutively with CmdT from the arabinose-inducible promoter in this experiment. So it is initially translated with CmdT, assists in folding and then is immediately turned over. Thus, while levels are not high and stable enough to be detected on a blot, CmdA is nevertheless initially available to help produce stable, active CmdT.

4. Figure 2E: A quantification, ideally of several independent experiments, would be desirable.

We now show three independent replicates of CmdA and CmdT levels pre and post infection (25 minutes) in Fig. S3c and indicate that there is a statistically significant difference.

5. Figure 2F: I assume the "WT" strain lacks CmdTAC? This would be consistent with the earlier data in the manuscript. If this is the case, can the figure be labelled more clearly? This applies to multiple figures in the manuscript.

We apologize for the confusion and have now updated the figure and figure legend. The figure shows two strain backgrounds, WT (MG1655) and $\Delta clpP$ (MG1655 $\Delta clpP$). For each strain background there is empty vector (–) and the *cmdTAC* plasmid (*cmdTAC*).

6. Figure 2: Can the order of panels be made more intuitive? Would it be worth to include loading controls for the western blots in the main figures throughout the manuscript? I realise

these are provided as part of the supplementary material. Where molecular weight markers are missing from figures, can these please be added?

The panels of figure 2 have now been rearranged to go in order and added molecular weight markers when appropriate. Due to space constraints, we prefer to leave loading controls in the supplemental material.

7. It would be reassuring to see a co-immunoprecipitation analysis of CmdC with Gp23 (and/or Gp21).

This is actually a difficult experiment to perform as, in our experience, epitope tagging Gp23 renders it non-functional. However, the IP-MS experiment in which we pull on an epitope tagged CmdC (see Fig. 3f), led to the clear co-precipitation of native Gp23 during a phage infection. We then bolstered these results by demonstrating (Fig. 3g) that expression of Gp23 along with its co-chaperonin Gp31 is sufficient, in the absence of any other phage components, to block growth of cells harboring CmdTAC. And we can confidently infer that CmdT gets liberated from CmdA and CmdC in this experiment as we detected ADP-ribosylation of RNA (see Fig. 4e) only when expressing Gp23, Gp31, and CmdTAC. Thus, we have multiple lines of evidence to support the notion that Gp23 is interacting with CmdC to drive CmdT activation.

8. p. 7, paragraph 4: "CmdTAC likely forms a stable complex in the absence of phage, with infection then triggering the ClpP-dependent degradation of CmdA, release of CmdT ..." - This hypothesis could also be tested by co-immunoprecipitation.

Although not explicitly highlighted in the text, we had previously shown that CmdTAC form a complex in the absence of phage infection through the CmdC IP-MS/MS results in Figure 3f where CmdA and CmdT co-precipitate with CmdC. We have additionally added as Figure 1h, and now state explicitly in the text (lines 120-123), results from a CmdT IP-MS/MS experiment in the absence of phage infection where CmdC and CmdA co-precipitate with CmdT.

9. Figure 3G: Is a growth curve for cmdTAC only available, for side-by-side comparison? I can see this has been done for Figure 1E, but a parallel control experiment for Figure 3G would have been helpful.

We have now moved the control experiment which includes *cmdTAC* only to the main figure for side-by-side comparison.

10. Figure 4A: Speculating about the identity of these bands may be secondary as the signal observed may be non-specific. Can a positive control (e.g., mono- and poly-ADP-ribose) be included?

It is likely that the bands on the left (uninfected cells) are non-specific because there is no known ADP-ribosyltransferase native to *E. coli* MG1655. However, on the right (infected cells), we see bands that likely result from the known T4 ARTs ModA, ModB, and Alt, essentially

serving as a positive control that the antibody is working. The text has been modified to indicate this more clearly (lines 235-239).

As a minor query, why did the authors choose this antibody rather than other ADP-ribose detection reagents (e.g., https://www.merckmillipore.com/GB/en/product/Anti-pan-ADP-ribose-binding-reagent,MM_NF-MABE1016)?

Thank you for the suggestion. Our lab had used the antibody reported in the paper with success in the past while studying DarTG. Others (e.g. Bullen et. al 2022) had also used this antibody successfully to study another bacterial ART. We do, however, appreciate the tip and will explore this other antibody in the future.

11. Figures 5B and C need molecular weight markers. It would be helpful to document mRNA ADP-ribosylation for Figure 5C in the supplementary material.

We have noted the molecular weight of DHFR in Fig. 5b-c. For Fig. 5c: importantly, CmdT is not added to the translation mix, so the only ADP-ribosylated species would be the CmdT-treated mRNA template. We now show in Fig. S5a the size shift of the CmdT-treated DHFR template indicating ADP-ribosylation.

12. Figure S2: In the PAE plots, can the authors highlight the domains and modules shown in panels A, B and C, for example by boxes? For the text (p. 5, paragraph 2), it may be worth pointing out that the extended C-terminus of CmdA is only predicted with low confidence, as apparent from the pLDDT score.

We have now updated Figure S2 to include boxes highlighting domain interactions in the PAE plot. We have additionally updated the text at line 119-120 to reflect the low confidence prediction of the CmdA C-terminus.

13. p. 3, paragraph 1: The term "poly-ADP-ribosyl polymerase" is outdated. Instead of "polymerase", we are encouraged to use the term "transferase", given that polymerases depend on templates whilst transferases do not.

We have now eliminated use of the term poly-ADP-ribosyl polymerase unless referring to a human protein with PARP in its name, e.g. PARP16. We now note this in the text when introducing ADP-ribosyltransferases (lines 29-31).

14. Details of mass spec data should be provided.

While we detailed the IP methodology in the methods section, we referenced another paper for details of the mass spec half of the IP-MS/MS as it was performed by the same core facility with an identical protocol. However, we recognize that this made for poor methods readability and have now rectified the issue. The methods section now includes a description of the on-bead reduction, trypsin digest, and LC-MS/MS done as part of the IP-MS/MS experiment (lines 955-

970). Additionally, while we had previously uploaded the raw data files from our mass spec analyses to massIVE under doi:10.25345/C52J68G0H for the IP-MS/MS data and doi:10.25345/C51N7XZ2Q for the ESI-MS/MS data we did not include any summary results. The IP-MS/MS deposit now includes a summary table of the search engine results which was available for that experiment.

15. RNA-DNA hybrids can also serve as substrates of reversible ADP-ribosylation. It would be interesting to examine the ADP-ribosylation of R-loop structures.

This is indeed an interesting experiment that we would like to pursue. We will note, however, that RNA:RNA duplexes are not modified (Fig. 6c). So, we suspect that RNA hybridized to DNA will also not be modified. Additionally, some cursory modeling using RosettaFoldNA suggests that duplexed RNA and more complex structures may not fit into the active site of CmdT.

16. What does the *alt-3* gene code for? Can a brief description of its function be added? Do the identified mutant variants represent loss or gain of function? What phenotype do the authors expect when *alt-3* is deleted? Non-experts in phage biology will have difficulties to follow.

Alt.-3 codes for a small protein of unknown function and with no detectable homology to proteins with clear function, as is relatively common with phage genes. The name derives from its location relative to the T4 gene *alt* (3 genes downstream). There is no phenotype that has been reported or that we have seen for an *alt.-3* deletion. We have added to the text to clarify this (line 194). Please also see our response to Reviewer 1, query 10 above where we provide additional data and commentary on *Alt.-3* and where we indicate that our preference is to not speculate in the text on *Alt.-3* function as it is somewhat ancillary to the paper's overall narrative, which is focused on CmdT activation and mechanism of action.

17. What portion of translation inhibition is due to host endoribonucleases and viral mRNA degradation?

Although the precise answer is unknown, we show in Fig. 5c that CmdT-treated mRNA is not translated in an in vitro translation system, which to our knowledge does not contain endoribonucleases.

18. Can the authors speculate why a phage defense system would be toxic to the bacterial cells themselves? Later data show that host mRNAs can also be ADP-ribosylated. Also, the double-edged sword of the defense system protecting the cells on the one hand and killing them on the other (in the experimental setting) could be introduced more explicitly, early on in the manuscript. Readers without an expert background in this field might get confused by the two sets of assays (viral plaque formation and viability).

We have expanded our introduction to the concept of “abortive infection” early on in the manuscript to clarify this point for readers less familiar with the phage defense field. In short,

abortive infection is a mechanism that leads to inhibition or death of the originally infected cell, but which protects the rest of the bacterial population from progeny viruses somewhat analogous to programmed cell death.

19. Do the authors have direct biochemical evidence that *cmdT** is catalytically inactive? Numerous ARTs featuring an HYE/D triad display poly-ADP-ribosyltransferase activity. I assume the *CmdT* is a mono-ART, but have the authors explored the nature of the modification more extensively?

Immuno-northern blots (see data panels directly below) indicated that *cmdT***A* produces no signal for ADP-ribosylation compared to *cmdTA*, supporting the notion that *CmdT** is inactive. We did not initially include this experiment as we had demonstrated that the *CmdT** mutation abolishes defense (Fig. S1d) and toxicity (Fig. 2a), but we are happy to include these data if the reviewer thinks they are essential.

Our *in vitro* experiments strongly suggest that *CmdT* is a mono-ART. We only see a single band for ADP-ribosylated oligos containing a single GA dinucleotide. However, more importantly, we note that no poly-ADP-ribosylated nucleotides were observed in our HPLC analysis shown in Fig. 6g and 6h.

20. Can the authors explore whether ADP-ribose hydrolysis reverses the effect of *cmdT*?

This is an interesting idea, but we are not aware of an enzyme known to catalyze the hydrolysis of ADP-ribose from RNA. At this stage we believe we have shown strong evidence of RNA ADP-ribosylation, but will keep this experiment in mind in the future.

21. Can the authors provide suggestions for how *CmdA* might block *CmdT* function?

Unlike some toxin-antitoxin systems, the structured domain of *CmdA* does not seem to occlude the active site of *CmdT*. We can only speculate that either *CmdA* binding leads to a

conformational shift in CmdT which disrupts the active site, or that some combination of the unstructured domain of CmdA and CmdC blocks CmdT's active site. We are actively trying to solve the crystal structure of the ternary complex and CmdT alone to definitively answer this question.

General comment: Can figure legends please be provided right underneath the figures? This would help the reviewers with assessing the work. Thank you!

Our apologies for not providing them underneath the figures and not including line numbers. These oversights have been corrected.

Reviewer Reports on the First Revision:

Referees' comments:

Referee #1 (Remarks to the Author):

Vassallo et al have provided a robust response to my comments, and those of the other reviewers. I congratulate the authors on an excellent study.

I have no other comments that need to be addressed.

Referee #2 (Remarks to the Author):

Personally, I think the manuscript would have benefitted from some additional mechanistic insight into how the modification prevents the mRNAs from entering translation, however, the authors argue that doing for example ribosome-seq is beyond the scope of the paper. I agree this is a lot of work, but one could also have performed more simply RNA-seq experiments looking at mRNA association with ribosomal fractions. Anyway, I guess its the editors choose as to what extent of mechanism insight is required for publication in Nature.

Referee #3 (Remarks to the Author):

I thank the authors for sufficiently addressing (nearly) all the points I had raised. Some remaining points:

- line 113: reference is missing

- In my original comment regarding molecular weight labels, I referred to labels for molecular weight markers (standards). Can these please be included wherever possible? Providing labels for the theoretical molecular weights of molecules of interest (e.g., Figure 5b) is less than ideal. Some panels still lack markers altogether.

- I encourage the authors to include the additional biochemical data included in response to point 19 of my original review. I think it is important to not only characterise cmdT* in terms of defense and toxicity but also biochemical activity.

- It is common practice to include spreadsheets with the list of interactors identified in IP-MS/MS and confidence criteria to enable non-specialised to assess the interactomes. Can the authors please include these details for IP-MS/MS data presented in Figures 1h and 3f?

Once again, I congratulate the authors on this very interesting work.

Author Rebuttals to First Revision:

Referees' comments:

Referee #1 (Remarks to the Author):

Vassallo et al have provided a robust response to my comments, and those of the other reviewers. I congratulate the authors on an excellent study.

I have no other comments that need to be addressed.

Referee #2 (Remarks to the Author):

Personally, I think the manuscript would have benefitted from some additional mechanistic insight into how the modification prevents the mRNAs from entering translation, however, the authors argue that doing for example ribosome-seq is beyond the scope of the paper. I agree this is a lot of work, but one could also have performed more simply RNA-seq experiments looking at mRNA association with ribosomal fractions. Anyway, I guess its the editors choose as to what extent of mechanism insight is required for publication in Nature.

That further work is needed to discern the exact mechanism of translation inhibition is now addressed in the Discussion section.

Referee #3 (Remarks to the Author):

I thank the authors for sufficiently addressing (nearly) all the points I had raised. Some remaining points:

- line 113: reference is missing

The missing reference has now been added.

- In my original comment regarding molecular weight labels, I referred to labels for molecular weight markers (standards). Can these please be included wherever possible? Providing labels for the theoretical molecular weights of molecules of interest (e.g., Figure 5b) is less than ideal. Some panels still lack markers altogether.

All blots and gels, including molecular weight markers can now be found in Supplemental Fig. 1. Note that for immunoblots, membranes were first stained and imaged to determine molecular weight marker positions and then probed and imaged separately by chemiluminescence. Images were aligned, as shown in Fig. S1, to relate chemiluminescent bands to molecular weight markers, as shown in main figures.

- I encourage the authors to include the additional biochemical data included in

response to point 19 of my original review. I think it is important to not only characterise cmdT* in terms of defense and toxicity but also biochemical activity.

The CmdT* biochemical data has now been incorporated into the paper as Extended Data Fig. 5a and is referenced in the text.

- It is common practice to include spreadsheets with the list of interactors identified in IP-MS/MS and confidence criteria to enable non-specialised to assess the interactomes. Can the authors please include these details for IP-MS/MS data presented in Figures 1h and 3f?

The high confidence interactors for both the CmdT and CmdC IP-MS/MS experiments have now been included in the paper as SI Table 1. Additionally, a more in depth spreadsheet is available at the massIVE deposit doi:10.25345/C52J68G0H.

Once again, I congratulate the authors on this very interesting work.